# Impact of secondary ice production on thunderstorm electrification under different aerosol conditions

Shiye Huang[1], Jing Yang[1,2,*], Jiaojiao Li[1], Qian Chen[1], Qilin Zhang[1], Fengxia Guo[1]

[1]Collaborative Innovation Center on Forecast and Evaluation of Meteorological Disasters (CIC-FEMD)/China Meteorological Administration Aerosol-Cloud and Precipitation Key Laboratory, Nanjing University of Information Science & Technology, Nanjing, 210044, China.
[2]China Meteorological Administration Key Laboratory of Cloud-Precipitation Physics and Weather Modification (CPML), Beijing, 100081, China.

*Correspondence to*: Jing Yang (jing.yang@nuist.edu.cn)

**Abstract.** Aerosol and secondary ice production (SIP) processes are both vital to charge separation in thunderstorms, but the relative importance of different SIP processes to electrification under different aerosol conditions is not well understood. In this study, using the Weather Research and Forecasting (WRF) model, we investigate the role of four different SIP processes in charge separation with different aerosol concentrations, including the rime-splintering (RS), the ice-ice collisional (IC) breakup, shattering of freezing drops (SD), and sublimational breakup (SK). It is found that as the aerosol concentration increases, more but smaller cloud droplets can be lofted to mixed-phase regions. The warm rain process is suppressed, and the declined raindrop concentration leads to fewer graupel particles. In a clean environment (aerosol concentration < 1000 cm$^{-3}$), the SD process is the most important to ice production between 0℃ and -10℃, while in a polluted environment (aerosol concentration ≥ 2000 cm$^{-3}$), the RS process contributes the most between 0℃ and -10℃. The IC process is important between -10℃ and -20℃. The SIP processes and the increase in aerosol concentration both enhance the noninductive charging rate. However, aerosol and SIP processes have opposite impacts on the charging reversal, which implies they play different roles in controlling the charge structure. In a clean (polluted) environment, the SD (RS) process has the greatest effect on the charge structure. Both the SIP processes and the increase in aerosol concentration enhance the electric field, but the enhancement of flash rate by increasing aerosol concentration is much greater than SIP.

## 1 Introduction

Thunderstorm, accompanied by lightning, is one of the most serious natural hazards to the public (Fierro et al., 2013). The charge structure of thunderstorms determines the frequency and intensity of lightning, and the charge separation within the clouds depends on the complicated dynamical and microphysical processes. The study of thunderstorms, especially their microphysics and electrification, has been a hot topic in meteorology for decades (Takahashi, 1983; Saunders et al., 1991; Saunders and Peck, 1998; Mansell et al., 2005). However, the impact of different microphysical processes on cloud electrification is not fully understood. The uncertainty in modelling cloud microphysics leads to biased lightning prediction in

numerical simulations. Among the various microphysical processes, aerosols and different ice crystal formation mechanisms play a crucial role (Pan et al., 2022; Phillips et al., 2020; Phillips and Patade, 2022; Yang et al., 2016; 2020).

Cloud condensation nuclei (CCN), which are aerosol particles capable of forming cloud droplets, play an important role in cloud microphysics. According to the Twomey effect, an increase in aerosol concentration leads to an increase in cloud droplet number concentration (Twomey, 1977). Due to water vapor competition, higher aerosol concentrations lead to smaller cloud sizes, causing smaller collision efficiencies. Collision coalescence is known as an essential process for warm-rain initiation. This means that the increase in aerosol concentration may suppresses the warm rain process, which is proven by observational studies (Rosenfeld, 1999; Rosenfeld, 2000; Rosenfeld and Woodley, 2000). In addition, aerosol also has significant impacts on the microphysics in mixed-phase region. The increase in aerosol concentration reduces cloud droplet size, thus more small droplets can be lofted to mixed-phase region (Rosenfeld, 1999; 2000), which may enhance the freezing of cloud droplets and facilitate cold rain process (Rosenfeld and Woodley, 2000; Rosenfeld et al., 2008; Hoose et al., 2010; Sherwood, 2002; Jiang et al., 2008). The increase in CCN condensation may also intensify the hail growth through riming in convective clouds (Chen e al., 2010), and the enhanced ice growth rate can produce more latent heat, which in turn invigorate the convections (Khain et al., 2005; Wang, 2005). The aerosol impact on microphysics will eventually modify the electrification of thunderstorms (Pan et al., 2022).

Cloud electrification is strongly controlled by the complicated ice production processes in thunderstorms. In mixed-phase region, the heterogeneous nucleation, which depends on ice nucleating particles (INP), provides the primary ice. However, it is found that the primary ice production (PIP) is insufficient to explain the observed high ice concentration in convective clouds (Hobbs, 1969; Hobbs and Rangno, 1985; Koenig, 1963). Secondary ice production (SIP) is recognized as a major contributor to the fast ice generation at temperatures warmer than the homogeneous freezing temperature (Hallett and Mossop, 1974; Mossop and Hallett, 1974; Santachiara et al., 2014; Korolev 2020). There are several different mechanisms of SIP proposed by previous studies, such as ice splintering during riming (hereafter RS, Hallet and Mossop, 1974), ice-ice collisional breakup (hereafter IC, Schwarzenboeck et al., 2009), shattering of drops during freezing (hereafter SD, Gagin, 1972; Pruppacher and Schlamp,1975; Kolomeychuk et al., 1975), and ice sublimation breakup (hereafter SK, Bacon et al., 1998; Dong et al., 1994; Oraltay and Hallett, 1989). The ice concentration produced by SIP processes is orders of magnitude higher than PIP, but PIP is a prerequisite for SIP, and changes in PIP (e.g., induced by increasing aerosol concentration) can affect the SIP processes (Sullivan et al., 2018). Based on numerical simulations, Qu et al. (2020) found that the number concentration of ice splinters produced by SD process increases as low-level CCN concentration increases from 100 $cm^{-3}$ to 1000 $cm^{-3}$, and decreases as CCN concentration increases from 1000 $cm^{-3}$ to 2000 $cm^{-3}$. Mansell and Ziegler (2013) showed that ice splinter production through the RS process increases remarkably as CCN concentration increases from 700 $cm^{-3}$ to 1500 $cm^{-3}$, and continues to increase until meeting a plateau above 3000 $cm^{-3}$. Tan et al. (2015) suggested that secondary ice produced by graupel collecting big cloud droplets (> 24 μm) under low CCN conditions is greater than that under high CCN conditions.

Since SIP processes enhance ice production, they will inevitably affect electrification in clouds with different CCN concentrations as collisional separation between ice crystals and graupel is the most important charging process (Mansell et al., 2005; Saunders et al., 1991; Saunders and Peck, 1998; Takahashi, 1978). Some previous studies have been conducted to investigate the impact of the RS process on electrification. Baker et al. (1995) find an inverted charge structure generated by

rime splintering based on numerical simulations. Lighezzolo et al. (2010) extended the rime growth experiments conducted by Hallett and Saunders (1979) and found the average charge of one ice splinter ejected from the graupel is -14 fC. Recently, studies of the impact of several other SIP processes on cloud electrification have been conducted. Phillips and Patade (2022) investigated the impacts of three SIP processes on deep convection using the Aerosol–Cloud model, and they found that the graupel–snow collisions account for the majority of the charge separated in the simulated storm, and the storm electrification

can be significantly altered by SIP from ice-ice collisional breakup (Phillips et al., 2020; Phillips and Patade, 2022). Huang et al. (2024) investigated the effects of three SIP mechanisms on cloud microphysics and electrification by a realistic simulation of a squall line and found that SIP processes significantly affect ice generation and charge separation. Without any SIP process, the storm had an inverted tripole structure, while with all the SIP processes implemented, the storm obtained a normal dipolar charge structure. Yang et al. (2024) simulated a cold-season thunderstorm in southeast China using the Weather Research and

Forecast (WRF) model containing four SIP processes, the results suggest that the RS process is the most important SIP process to cloud electrification in their case.

     Previous studies have demonstrated that both CCN and SIP processes are vital for thunderstorm electrification, but till now, to our best knowledge, no study has explored the relative importance of different SIP processes under different CCN

concentrations in thunderstorms. Observations and simulations have merely confirmed the strong link between aerosols and lightning activity through ice-phase particles but do not consider the various SIP processes (Chaudhuri and Middey, 2013; Khain et al., 2008; Lynn et al., 2020; Naccarato et al., 2003; Pinto et al., 2004; Shukla et al., 2022; Stallins et al., 2013; Tan et al., 2015; Wang et al., 2011). In this paper, the parametrizations of four SIP mechanisms, noninductive and inductive charging schemes as well as a bulk discharge model are implemented in WRF model with spectral bin microphysics scheme to simulate

a squall line that occurred in Southeast China on 29-30 May 2022. Sensitivity experiments are made using different CCN concentrations to study the impacts of SIP processes on cloud electrification under clean and polluted conditions. The SIP processes considered in this paper are RS, IC, SD, and SK processes. The rest of this work is structured as follows: Sect. 2 introduces the model setup for simulating the squall line. The results are displayed in Sect. 3. Sect.4 discusses and summarizes this study, respectively.

## 2 Case description and design of numerical experiments

### 2.1 Case description

The simulated mesoscale convective system (MCS) is a squall line that occurred in Southeast China from May 29th to May 30th, 2022 (unless otherwise noted, UTC time is used throughout the remainder of this paper). The geopotential height, temperature as well as wind field of 500 hPa and 850 hPa are shown in Figure 1, which is plotted using the NCEP FNL reanalysis data. A deep trough occurred at 500 hPa and baroclinicity was present at 850 hPa (Fig. 1b and d). At 850 hPa (Fig. 1d), southwest winds prevailed at latitudes lower than 28°N, while northeast winds prevailed at latitudes higher than 28°N, suggesting the presence of low-level convergence. In addition, the humid conditions at low levels and the dry conditions at high levels (Fig. 1a and b) are suitable for the formation of convection. The squall line investigated here is the same as that in Huang et al. (2024), which confirmed SIP has a strong impact on the charge structure for this case with a high aerosol concentration, but it is not known which SIP process is more important under different aerosol conditions.

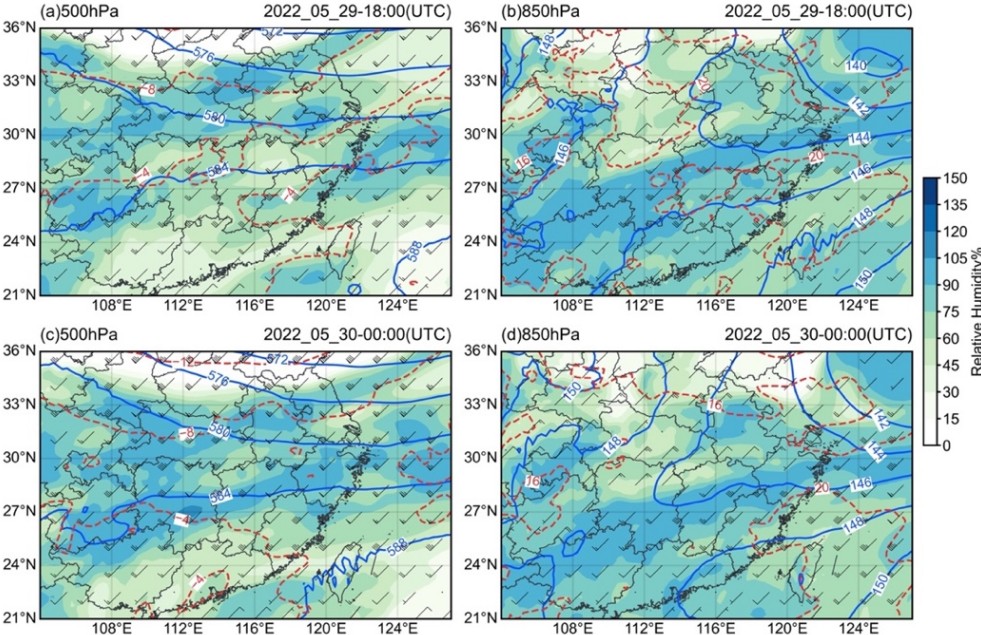

**Figure 1. The Geopotential height (blue line), temperature (red line), and wind (black arrow) at 500hPa (a, c) and 850hPa (b, d). The shaded background represents relative humidity.**

### 2.2 Description of model simulations

#### 2.2.1 Cloud microphysics

The microphysics scheme used in this paper is the fast version of spectral bin microphysical (Fast-SBM) scheme developed by Hebrew University Cloud Model (HUCM) (Khain et al., 2000). The Fast-SBM scheme describes the size distributions of

three types of hydrometeors: water droplets (cloud droplets and rain drops), low density solid particles (ice/snow), and high density solid particles (graupel/hail). In contrast to the bulk microphysics scheme, this scheme uses 33 double mass bins to characterize the particle size distribution. In previous studies, the effectiveness of the SBM and bulk schemes for simulating cloud microphysics is compared, and many studies suggested that the SBM scheme provides simulation results that are closer to real observations (Khain et al., 2009, 2015). In Fast-SBM scheme, the number concentration of CCN is calculated using an empirical formula (Pruppacher and Klett, 1997), the Twomey equation, $N_{CCN} = N_0 S_w{}^k$, where $S_w$ refers to supersaturation with respect to water, $N_0$ represents the CCN concentrations at 1% supersaturation, and $k$ is the slope of the CCN size distribution. In this study, we use 0.4 for $k$, which is the default value in the model.

The default version of the fast-SBM scheme has different mechanisms of PIP, including deposition/condensation nucleation, contact nucleation (Meyers et al., 1992), and immersion freezing (Vali, 1975). However, only a single SIP mechanism is used, which is the RS process. In the present study, we implement three other mechanisms of SIP (IC, SD, and SK) in the model. The parameterization of the rime-splintering process is derived from two classical laboratory studies (Hallett and Mossop, 1974; Mossop and Hallett, 1974). Fragment production occurs between -3°C and -8°C and reaches the maximum at -5°C. At -5°C, one ice fragment is generated for every 350 droplets collected by a graupel particle through the rime process. Phillips et al. (2017) have developed a theoretical formulation to describe secondary ice fragments produced during ice-ice collision based on an energy conservation principle and estimated theoretically uncertain parameters depending on laboratory and field experiments (Takahashi et al., 1995; Vardiman, 1978). The production rate of ice splinters is related to particle size, rimed fraction, ice habits and temperature. In this study, the selection of ice habit is based on temperature according to Phillips et al. (2017) and the rimed fraction is 0.2. Phillips et al. (2018) compiled previous laboratory studies and proposed a two-mode formulation to describe the fragmentation of freezing drops. The first mode of this fragmentation of freezing drops mechanism represents the collision between frozen drops and smaller ice particles and produces both big and tiny splinters. The second mode refers to the collision between frozen drops and bigger ice particles and produces tiny splinters. The first bin of Fast-SBM mass bins is used to represent the mass of tiny fragments. The number and size of ice splinters depend on parent drop size and environmental temperature, as well as collision energy. Deshmukh et al. (2022) proposed a formula to describe the number of ice fragments during the SK process that applies to dendritic ice crystals and heavily rimed particles (e.g., graupel). The size of parent ice particles and the relative humidity on ice control the number of ice fragments. In this study, we only consider the SK process of ice/snow. The equations and implementation of the four SIP processes in the Fast-SBM scheme are described in detail by Yang et al. (2024).

**2.2.2 Cloud electrification**

It is known that the noninductive charge mechanism, referring to the rebounding collisions between graupel particles and ice crystals, is the primarily important charging mechanism in thunderstorms (Brooks et al., 1997; Mitzeva et al., 2006; Saunders et al., 1991; Saunders and Peck, 1998; Takahashi, 1978). The noninductive charging parametrization developed by Saunders

and Peck (1998, hereafter SP98) based on the rime accretion rate (RAR) is implemented in this paper as in many previous studies (Fierro and Mansell, 2017; Fierro and Reisner, 2011; Huang et al., 2024; Yang et al., 2024). The charge transfer in this scheme is determined by three terms: 1) charge transferred during each collision between graupel and ice ($\delta q_{gi}$); 2) collision kernel between graupel and ice; 3) concentration of graupel and ice. $\delta q_{gi}$ is determined by RAR, which is a function of liquid water content (LWC) and terminal velocity of graupel. The noninductive charging occurs where RAR is greater than 0.1 g m$^{-2}$ s$^{-1}$ (Mansell et al., 2005; Saunders and Peck, 1998). The negative $\delta q_{gi}$ denotes that an RAR is smaller than the critical RAR, and the positive $\delta q_{gi}$ denotes an RAR greater than the critical RAR. The critical RAR is a function of temperature. Besides, an inductive charging parameterization (Mansell et al., 2005) is also implemented in the SBM. Under the action of an electric field, the particles polarize through ions of opposite signs accumulating on opposite sides, and charge transfer may occur between particles when they collide and separate. The inductive charging rate is mainly related to collision efficiency between graupel and droplet, rebound probability, concentration of graupel and droplet, as well as vertical electric field. To simulate the discharge, a bulk scheme is used, in which a 30% charge is released at points where the electric field exceeds a threshold. This scheme cannot describe elaborate discharge channels (Fierro et al., 2013). Although more sophisticated lightning models have been developed in previous studies, given the computational cost and the desire to relate the lightning rate to cloud microphysics properties, a bulk discharge scheme is appropriate for this study.

### 2.2.3 Model setup

The real-case mode of WRF is employed to simulate the squall line. The simulated domains are shown in Figure 2, and the resolution of these two nested domains are 9 km and 3 km, respectively. The number of vertical levels is 61, and the top pressure is 50 hPa for the two domains. The initial field and boundary conditions from 06:00 UTC on 29 May were provided using FNL reanalysis data with 0.25° × 0.25° resolution. The simulation starts at 06:00 UTC on 29 May 2022 and runs for 24 hours. The period lasting from 16:00 UTC on 29 May 2022 to 06:00 UTC on 30 May 2022 is of interest.

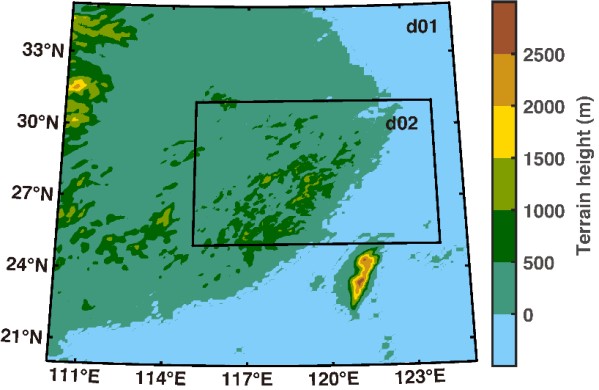

**Figure 2. Terrain map of the model domain.**

In the present study, the Kain-Fritsch cumulus scheme is activated in the outer domain while turned off in the inner domain. The shortwave and longwave radiation are both parametrized using the Rapid Radiative Transfer Model (RRTM) (Mlawer et al., 1997). The Yonsei University planetary boundary layer scheme (Hong et al., 2006), the Revised MM5 surface layer scheme (Jiménez et al., 2012), and the Noah Land Surface Model (Tewari et al., 2004) are also employed in this paper.

Sensitivity experiments are conducted in two main aspects, on the one hand by activating different SIP processes and on the other hand by applying different CCN concentrations. To explore the significance of CCN concentrations for cloud microphysics and electrification, simulations are carried out with different settings of the coefficient $N_0$ in the Twomey equation: 400, 1000, 2000, and 4000 cm$^{-3}$. To investigate the impact of SIP, we conduct simulations with or without SIP processes. The design of the sensitivity experiments is shown in Table 1. The first part of the experiment name indicates activation of SIP: "noSIP" denotes that none of the SIP mechanisms is considered, and "4SIP" means four SIP mechanisms are implemented in simulation. The number in each experiment name indicates the value of $N_0$. This squall line occurred in southeast China, where the aerosol concentration is typically high, Qu et al. (2017) suggested an $N_0$=4000 cm$^{-3}$ for this region, and this value is also used in Huang et al. (2004), therefore, in this paper, the 4SIP-4000 experiment is set as a control experiment.

**Table 1. Description of the 8 sensitivity experiments.**

| Experiment | $N_0$ | SIP processes |
|---|---|---|
| noSIP-400 | 400 | \ |
| noSIP-1000 | 1000 | \ |
| noSIP-2000 | 2000 | \ |
| noSIP-4000 | 4000 | \ |
| 4SIP-400 | 400 | RS, IC, SD, and SK |
| 4SIP-1000 | 1000 | RS, IC, SD, and SK |
| 4SIP-2000 | 2000 | RS, IC, SD, and SK |
| 4SIP-4000 (control) | 4000 | RS, IC, SD, and SK |

### 2.3 Observational datasets

In this study, we use observed radar reflectivity and lightning locations measured by satellite to evaluate the model. The observed radar data are derived from the grid products provided by 32 S-band radars located across southeast China. Each radar has a detection radius of 230 km, a resolution of 250 m, and a 1°-beam-width. The radar takes 6 minutes to complete a volume scan containing 9 elevation angles (0.5, 1.5, 2.4, 3.4, 4.3, 6.0, 9.9, 14.6, and 19.5°).

The observed lightning location data were collected by the Lightning Mapping Imager (LMI), which is mounted on the second-generation Chinese geostationary meteorological satellite FY-4A to continuously detect lightning in China and its neighbouring areas. The LMI sensor consists of two 400 × 600 charge-coupled devices (CCDs) that can capture the lightning

optical radiant energy with a 2 ms pixel integration. When the CCD detects a signal, a real-time event processor filters the background noise using a multi-frame average background noise estimate. The average background noise from the frames preceding the current frame decides the filtering threshold. Considering the large background noise due to the reflection of sunlight by clouds and the atmosphere during the day, the LMI filtering threshold is dynamic. After filtering, possible lightning "events" are selected and then sent to the ground station in real-time. After removing false lightning signals and clusters, the lightning location product dataset is generated (Sun et al., 2021). Additionally, the brightness temperature captured by the FY-4A satellite is used to delineate the area of the deep convective cloud.

## 3 Results

### 3.1 Model validation

The radar reflectivity modelled by the 4SIP-4000 experiment is shown and compared with observations in Figure 3. As shown in the figure, this squall line was oriented southwest-northeast. The reflectivity in the strong convective core was approximately 55 dBZ. The storm moved towards the east with time, and dissipated after 06:00 UTC, May 30[th] (not shown). Although the simulated radar reflectivity deviates to some extent from observations, the simulation results well reproduce the macro-morphology, the occurrence location, and the eastward tendency of this squall line. A comparison of the simulated results and observations  reveals that the mean absolute errors of modelled reflectivity at 00:00,02:00 and 04:00 are 12 dBZ, 11 dBZ and 12 dBZ, respectively. The area where the radar reflectivity bias is smaller than 15 dBZ are 67%, 71%, and 67% at 00:00, 02:00 and 04:00, respectively. Our sensitivity tests show the 4SIP-4000 experiment is more consistent with observation than the other experiments, in which the modelled reflectivities are higher (shown later in Fig. 5).

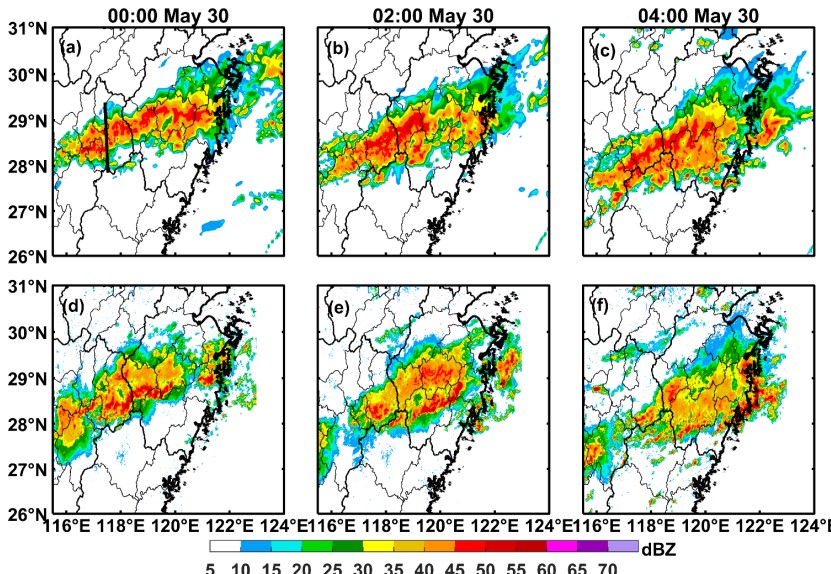

**Figure 3. The (a-c) simulated and (d-f) observed radar reflectivity at 00:00, 02:00, and 04:00 on 30th May. The vertical black line in (a) shows the cross-section used in the subsequent investigation. The simulated results are from the 4SIP-4000 experiment.**

The modelled and observed locations of lightning events are compared in Fig. 4. According to the figure, it is seen that the simulated and observed lightning locations are in good agreement with each other and are both in the low-brightness temperature region, which characterizes the presence of strong convection. In general, the fact that the radar reflectivity and lightning locations are reasonably well simulated gives us the confidence to study the effects of the SIP processes on cloud microphysics and electrification processes under different aerosol conditions.

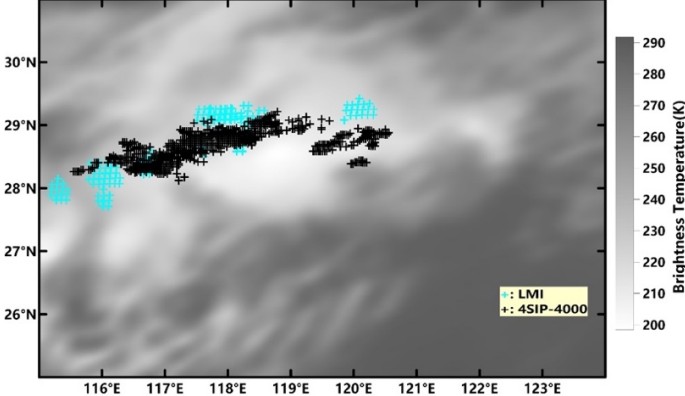

**Figure 4. The simulated and observed lightning locations during the storm. The shaded field indicates brightness temperature. The simulated results are from the 4SIP-4000 experiment.**

## 3.2 Impacts of SIP on cloud microphysics with different CCN concentrations

The impacts of aerosol and SIP processes on cloud microphysics in different sensitivity studies are analyzed in this section. Figure 5 illustrates the modelled radar reflectivity variation with time and altitude. It shows that the simulated reflectivity

decreases with the addition of the SIP process. As the CCN concentration increases, the simulated reflectivity increases in the noSIP experiments and decreases in the 4SIP experiments. According to the model validation in Section 3.1, it is suggested that the three experiments other than 4SIP-4000 overestimate the reflectivity even more. Since the modelled reflectivity is estimated for a wavelength of 10 cm, which falls to the Rayleigh scattering regime, the magnitude is mainly controlled by particle size rather than number concentration. Therefore, the particle size may be increased by increasing CCN concentration

from 400 to 4000 $cm^{-3}$ if no SIP process is considered, while the SIP processes reduce the particle size, indicating the aerosol and SIP play different roles in altering the microphysical properties.

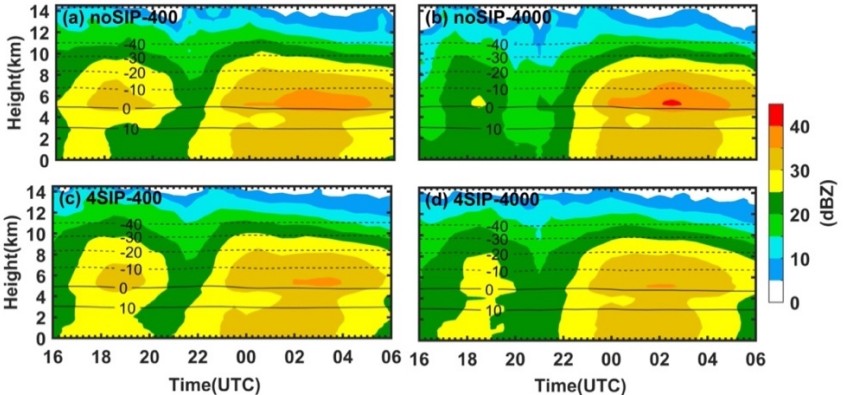

**Figure 5. Time-height evolution of the simulated reflectivity from (a) noSIP-400, (b) noSIP-4000, (c) 4SIP-400, and (d) 4SIP-4000 experiments.**


Figure 6 shows the temporal evolution of the mixing ratios of graupel, ice/snow, rain, and cloud droplets for noSIP-400, noSIP-4000, 4SIP-400, and 4SIP-4000 experiments. The results are averaged over the cloud at different heights, and the "in-cloud" condition is defined as the liquid/ice mixing ratio greater than $10^{-6}$ g $kg^{-1}$. As shown in the figure, more CCN results in a higher cloud droplet mixing ratio (Fig. 6a-d). Regardless of SIP processes, the maximum mixing ratio of cloud droplets with $N_0$=4000

$cm^{-3}$ is about 1.5 larger than that with $N_0$=400 $cm^{-3}$. Additionally, when $N_0$=4000 $cm^{-3}$, more cloud droplets could be lifted to higher altitudes, leading to more cloud droplets above the freezing level (Fig. 6a-d). In contrast, a higher concentration of CCN leads to a lower mixing ratio of raindrops (Fig. 6e-h) due to a less efficient warm rain process (Rosenfeld et al., 2008), and the graupel/hail mixing ratio is reduced as fewer raindrops are available for freezing (Fig. 6i-l). It is also noted that the ice/snow mixing ratio is slightly reduced by a higher CCN concentration, this is different from some previous studies which show more

CCN can enhance ice concentration in convective clouds as more cloud droplets are available above the freezing level for

freezing (Khain et al., 2008; Lynn et al., 2007). The reason is that the updraft strength is weakened and the total water mixing ratio is reduced by higher CCN concentration in the present case, and will be discussed in more detail later in Figure 9.

The addition of SIP processes increases the mixing ratios of graupel/hail significantly (Fig. 6i-l). In the experiments with $N_0$=4000 cm$^{-3}$, the increase in the mixing ratio of graupel/hail induced by the SIP processes is greater than in a clean environment ($N_0$=400 cm$^{-3}$). In addition, the mixing ratio of ice/snow is also enhanced due to the addition of SIP processes (Fig. 6m-p), but the rainwater mixing ratio is not significantly affected (Fig. 6e and g). For $N_0$=4000 cm$^{-3}$, the addition of SIP processes results in less liquid water mixing ratio above the freezing level (Fig. 6b, f, d, and h), probably due to the enhanced ice and graupel production and water depletion.

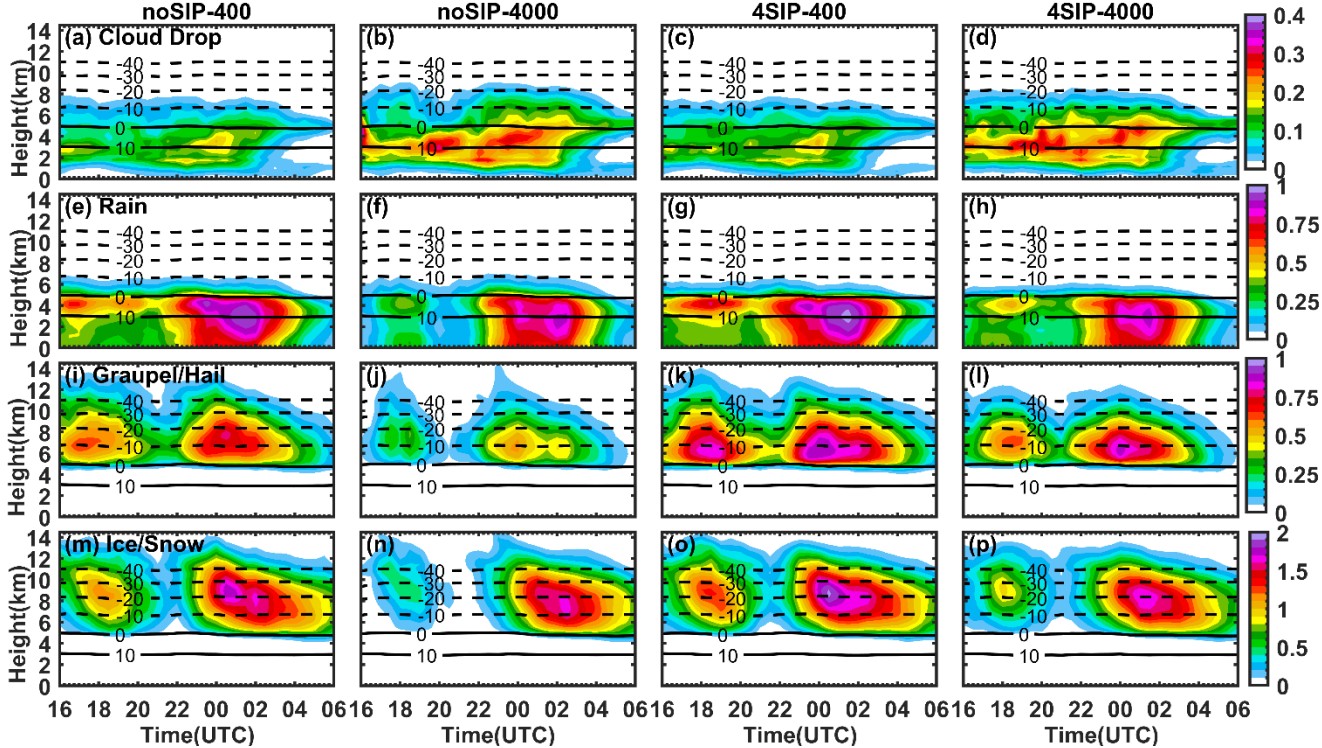

**Figure 6. Time-height evolution of the mean mixing ratio (unit is g/kg) of four particles for four sensitivity experiments. (a-d) cloud drop, (e-h) rain, (i-l) graupel/hail, (m-p) ice/snow. The first to fourth columns represent the results from noSIP-400, noSIP-4000, 4SIP-400 and 4SIP-4000, respectively.**

The vertical profiles of mean number concentration and mean diameter of graupel/hail, snow/ice, rain, and cloud drops from noSIP-400, noSIP-4000, 4SIP-400, and 4SIP-4000 experiments are shown in Figure 7. As expected, more aerosols lead to a noticeable increase in the concentration of cloud droplets (Fig. 7a) but a substantial decrease in size (Fig. 7e). In addition, the rain concentration significantly decreases in the experiments with $N_0$=4000 cm$^{-3}$ (Fig. 7b). However, the raindrop size increases

as the CCN concentration increases (Fig. 7f). At temperatures warmer than -20°C, more aerosols lead to less but bigger graupel
and snow particles without considering SIP processes (Fig. 7c and d).

The impacts of SIP processes are significant for graupel and snow while minor for liquid drops, this is found for experiments
with different CCN concentrations. With the implementation of SIP processes, the graupel concentration increases significantly
(Fig. 7c), while the graupel size decreases in the region warmer than -20 °C (Fig. 7g). The increases in snow concentration by
SIP processes are mainly found at temperatures between 0 °C and -20 °C (Fig. 7d). Such increase is more profound for $N_0$=4000
cm$^{-3}$, resulting in a bimodal structure with an additional peak at -8 °C (Fig. 7d). This is because, for high concentrations of
CCN, the RS process can significantly enhance ice production between 0°C and -10°C, while it is less efficient in a clean
environment ($N_0$ is less than 1000 cm$^{-3}$). We will demonstrate this statement later in Fig. 10.

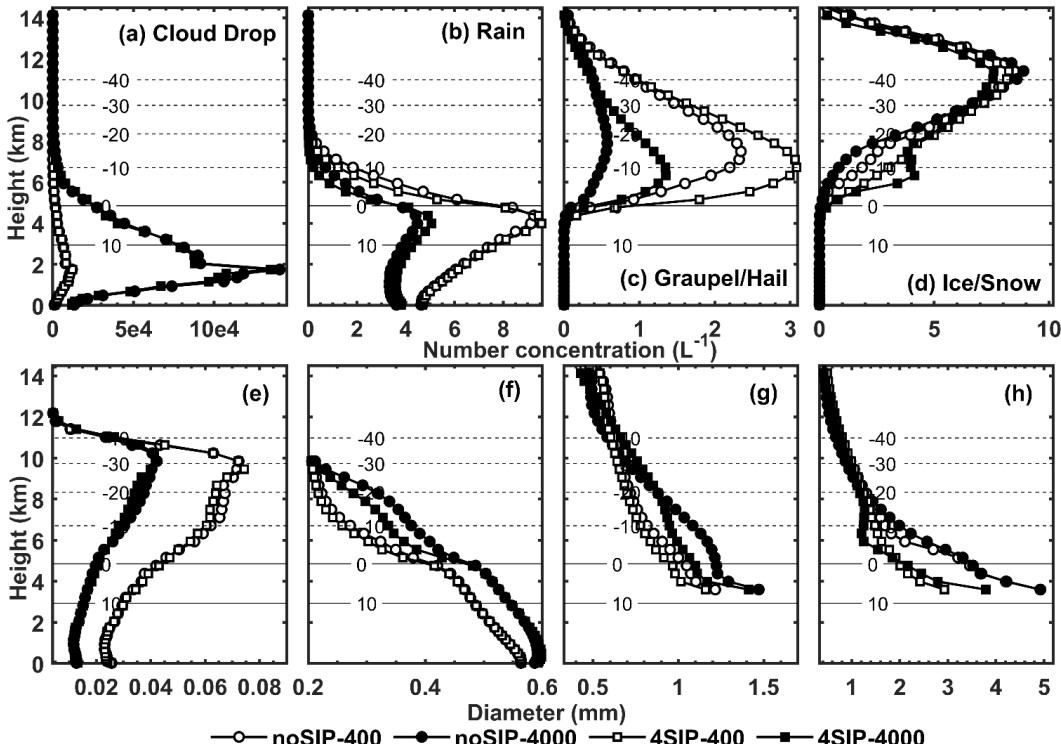

**Figure 7. The vertical profiles of mean concentration (upper row) and mean diameter (lower row) of (a, e) cloud drop,**
**(b, f) rain, (c, g) graupel/hail and (d, h) ice/snow from noSIP-400, noSIP-4000, 4SIP-400 and 4SIP-4000 experiment.**

The average particle number concentrations and mixing ratios obtained from noSIP-400, noSIP-4000, 4SIP-400, and 4SIP-
4000 experiments are summarized in Figure 8, which well illustrates the overall impacts of aerosol and SIP on cloud
microphysics. In general, as $N_0$ increases from 400 cm$^{-3}$ to 4000 cm$^{-3}$, the mixing ratio and number concentration of cloud
droplets increase (Fig. 8a and b), while the mixing ratio and concentration of graupel and rain decrease (Fig. 8c-f). Without

any SIP process considered, the ice/snow mixing ratio and concentration decrease as $N_0$ increases from 400 $cm^{-3}$ to 1000 $cm^{-3}$, suggesting weakened ice nucleation. Although the cloud droplet concentration is higher in noSIP-1000 than in the noSIP-400 experiment, the droplets that can be lifted to upper levels are insufficient to provide a higher ice concentration. In addition, the total water (liquid and ice) mixing ratio above the freezing level is lower in the noSIP-1000 experiment than in the noSIP-400 experiment. This finding implies an inhibited convection by increasing CCN concentration, which is demonstrated in Figure 9. It is seen that the mean updraft strength is relatively weak when $N_0$=1000 $cm^{-3}$, resulting in a lower total water mixing ratio. The decreased latent heat release due to less drop freezing is the plausible explanation for the weakened updrafts as $N_0$ increases from 400 $cm^{-3}$ to 1000 $cm^{-3}$. As $N_0$ increases from 1000 $cm^{-3}$ to 4000 $cm^{-3}$, the updraft velocity varies non-monotonically with increasing aerosol concentration in noSIP experiments, we do not see significant updraft enhancement and increase in total water mixing ratio, but the ice/snow concentration is enhanced (Fig. 8h), suggesting the aerosols-impact on microphysics exceeds its impact on dynamics if the CCN concentration is very high. We also investigated the maximum vertical velocity and obtained a similar conclusion (not shown). However, the ice/snow mixing ratio is relatively low when $N_0$=4000 $cm^{-3}$ though its concentration is high. This is because more ice crystals compete for the limited water vapor and liquid water, leading to a lower ice growth rate. Some previous studies also found this phenomenon in different case studies. Tan et al. (2015) noted that water vapor competition may result in a decrease in the ice crystal mixing ratio at aerosol concentrations between 1000 and 3000 $cm^{-3}$. Qu et al. (2017) pointed out that fewer snow crystals form in the simulation with more aerosols due to the declined depositional growth.

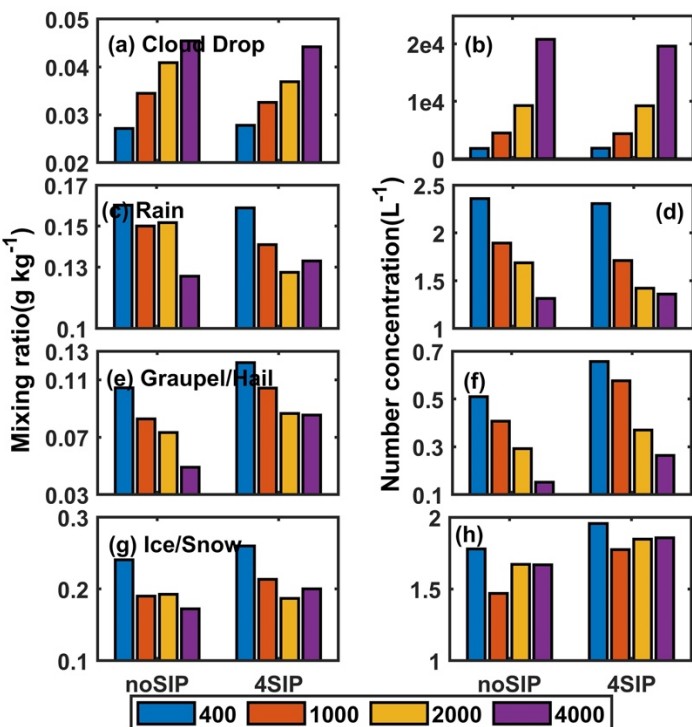

Figure 8. The time-domain-averaged mixing ratios (a, c, e, g) and number concentrations (b, d, f, h) of (a, b) cloud drop, (c, d) rain, (e, f) graupel/hail and (g, h) ice/snow from 8 sensitivity experiments.

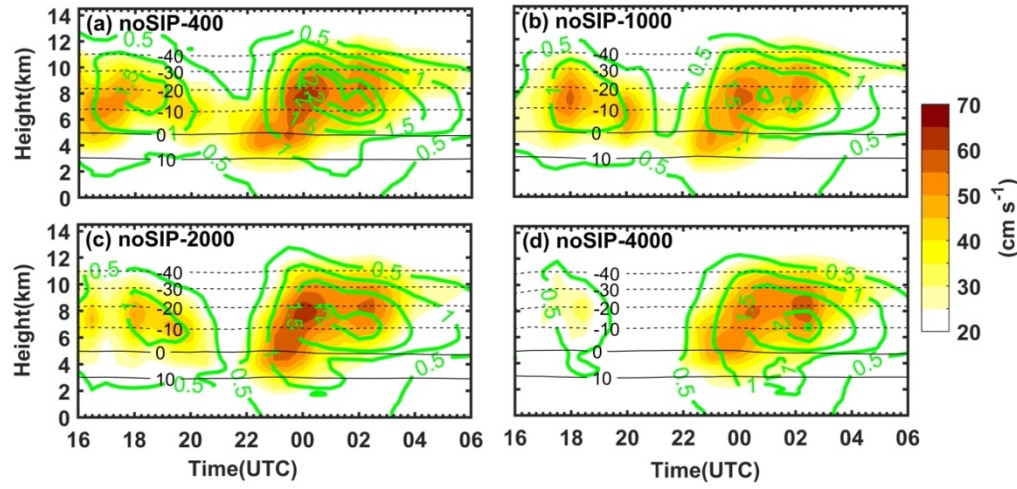

Figure 9. Time-height evolution of mean vertical wind velocity (cm/s, shaded) of (a) noSIP-400, (b)noSIP-1000, (c)noSIP-2000, (d)noSIP-4000. Green contour lines show all hydrometeors mixing ratio of 0.5, 1.0, 1.5, 2.0 and 2.2 g/kg.

The relative importance of the four SIP processes varies with different CCN concentrations. This can be illustrated by the production rates of secondary ice (Figure 10). According to Fig. 10, when $N_0$ is 400 cm$^{-3}$, SD produces the most secondary ice between 0 °C and -10°C (Fig. 10i), and IC produces the most secondary ice between -10 °C and -20 °C (Fig. 10e). As suggested by Von Terzi et al. (2022) and Georgakaki et al. (2024), collision of dendritic ice crystals is the reason for enhanced IC process between -10°C and -20°C. In this paper, the greater ice number concentration between -10°C and -20°C favors the ice collision, while smaller size of ice above -20°C isotherm is unfavorable for the collision between ice. When $N_0$ is 4000 cm$^{-3}$, the RS process has the highest secondary ice production rate between 0 °C and -10 °C (Fig. 10d).

The SIP processes influence both the graupel and ice concentrations. For graupel, which is mainly found at temperatures greater than -20 °C, SD is the most important SIP mechanism when $N_0$=400 cm$^{-3}$ and $N_0$=1000 cm$^{-3}$, while as $N_0$ increases to 4000 cm$^{-3}$ the RS process contributes the most to the enhancement of graupel concentration. This is confirmed in sensitivity tests using individual SIP processes (not shown), and in Fig. 10, which shows the ice production rate through the RS process increases with increasing CCN concentration (Fig. 10a-d), while the SD is less significant when $N_0$=4000 cm$^{-3}$ compared to $N_0$=400 cm$^{-3}$ and $N_0$=1000 cm$^{-3}$ (Fig. 10i-l). The IC process seems to have a strong enhancing effect on ice concentration, but it is efficient at relatively cold temperatures (between -10°C and -20 °C, Fig. 10e-h), and it has minor impacts on the graupel regardless of different values of $N_0$ (not shown). The size of secondary ice particles produced through IC is small, so the mixing ratio of ice is not significantly enhanced (not shown). The SK process occurs between -10 °C and -20 °C, but its ice production rate is orders of magnitude lower than the others (Fig. 10m-p), thus, it has minor impacts on the microphysics in the present case.

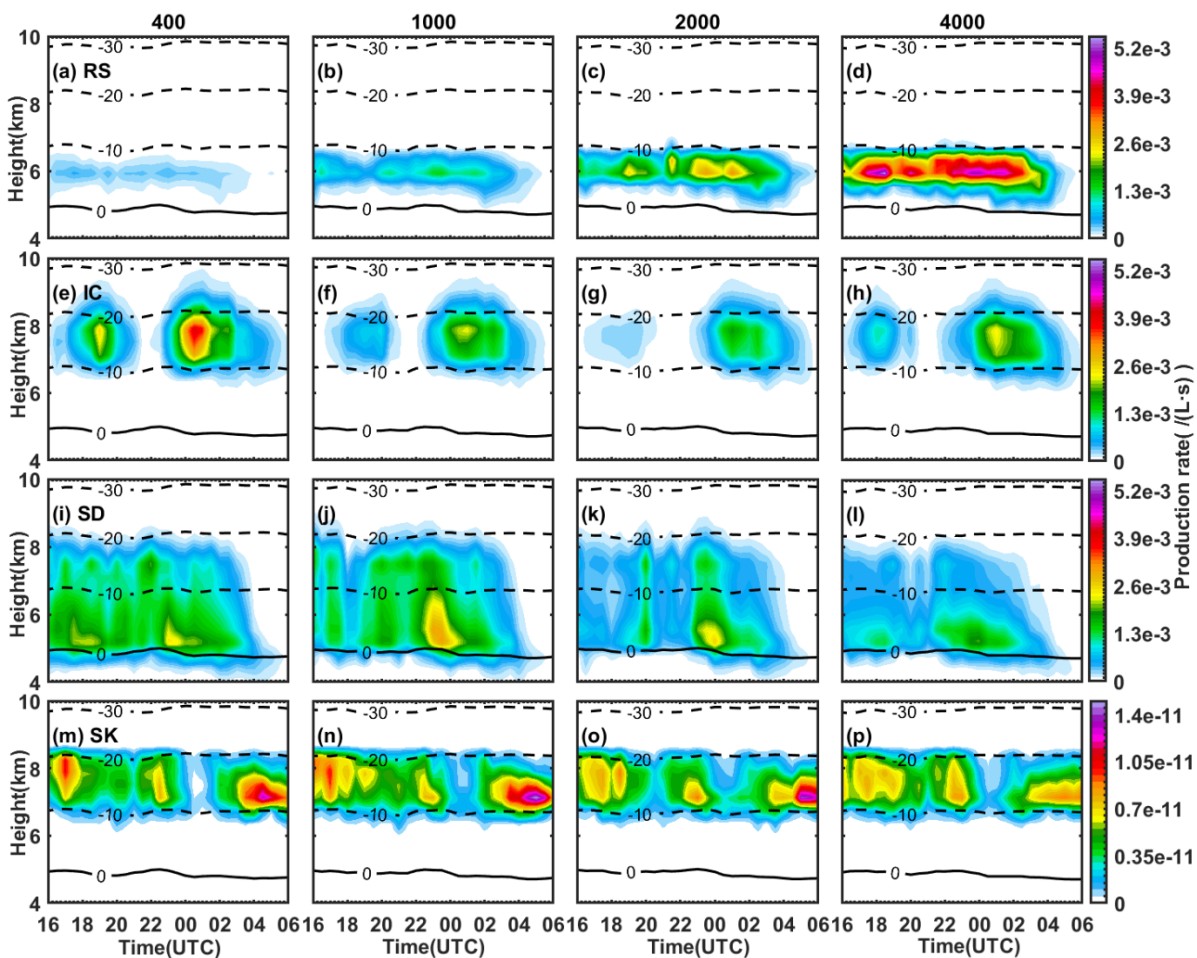

**Figure 10. Time-height evolution of secondary ice production rate of four mechanisms of SIP. (a-d) rime-splintering process (RS); (e-h) ice-ice collisional breakup (IC); (i-l) freezing drop shattering (SD) and (m-p) sublimational breakup (SK). The first to fourth column indicate $N_0$=400, 1000, 2000, 4000 cm$^{-3}$, respectively.**


To sum up, in the present case, the major impacts of aerosol and SIP processes on cloud microphysics are as follows, and we will show that they have substantial influences on cloud electrification.

1) More cloud droplets can be lifted to higher levels and the warm rain is suppressed as the CCN concentration increases.

2) Fewer graupel particles can form due to fewer rain-size drops for freezing as the CCN concentration increases.

3) Without any SIP process, the convective updraft is inhibited as $N_0$ increases from 400 cm$^{-3}$ to 1000 cm$^{-3}$, leading to a lower total water mixing ratio and a lower ice concentration. However, the aerosol impact on ice microphysics exceeds its impact on dynamics as $N_0$ increases from 1000 cm$^{-3}$ to 4000 cm$^{-3}$.

4) In a clean environment ($N_0 \leq 1000$ cm$^{-3}$), SD is the most important SIP mechanism to ice production between 0°C and -10°C, and the graupel concentration can be significantly enhanced.

5) In a polluted environment ($N_0 \geq 2000$ cm$^{-3}$), RS contributes the most to the graupel and ice production between 0°C and -10°C. IC contributes the most to ice production between -10°C and -20°C.

**3.3 Impacts of SIP on cloud electrification with different CCN concentration**

Cloud electrification is associated with the collisions between ice/snow and graupel and the interaction between droplets and
graupel. Changes in cloud microphysics by aerosols and SIP processes can result in changes in thunderstorm charge structures. Figure 11 shows the temporal evolution of the noninductive and inductive charging rates. In general, the noninductive charging rate is significantly greater than the inductive charging rate, which is consistent with the opinion that the noninductive charging process is the main charging process of thunderstorms (Reynolds et al., 1957; Saunders et al., 1991; Takahashi, 1978, 1983). The noninductive charging rate has a distinct dipole structure with upper negative and lower positive regions (Fig. 11a-d). The
relatively low LWC and the small size of graupel in the upper region lead to a smaller RAR, which is more likely to be less than the critical RAR and result in a negative noninductive charging rate. This is also found in previous studies, for example, Fierro and Mansell (2017) simulated idealized tropical cyclones and designed sensitive experiments to investigate the impact of the wind shear and sea surface temperature on cloud microphysics and electrification. They found that positive charging occurs between 6.5 and 8 km with negative charging above that over a deeper layer, between 8 and 11 km. Implementation of
the four SIP processes as well as the increase in aerosol concentration causes an enhancement of the noninductive charging rate. It is noted that the reversal temperatures of noninductive charging rates are about -8°C in the noSIP-400 experiment, -18°C in the noSIP-4000 experiment, -5°C in the 4SIP-400 experiment, and -15°C in the 4SIP-4000 experiment, respectively (Fig. 11a-d), indicating aerosol and SIP processes have opposite impacts on the charging reversal: higher CCN concentration results in a colder reversal temperature, while SIP processes lower the reversal level. In experiments with larger $N_0$, the reversal
temperature position is elevated because more aerosols lead to more cloud droplets above the freezing level. The addition of four SIP processes enhances the ice concentration and leads to less LWC (Fig. 6 and 7), which consequently results in a warmer reversal temperature. The inductive charging rate is much smaller than the noninductive charging rate, but it cannot be ignored. Similar to the noninductive charging rate, both the increase in CCN concentration and SIP processes can enhance the inductive charging rate (Fig. 11e-h). The inductive charging rate in the 4SIP-4000 experiment is the greatest among the four experiments.
Since the impact of sublimational breakup of ice is minor, the modelled charging rate using four SIP processes is comparable to that simulated using three SIP processes (without SK) in Huang et al. (2024).

Statistically, as $N_0$ increases from 400 cm$^{-3}$ to 4000 cm$^{-3}$, the noninductive charging strengthens, with the mean positive noninductive rate and negative rate increasing by 4.9 and 3.2 times, respectively. The inductive charging rate increases too,
with positive and negative rates increasing by a factor of 8.7 and 4.9, respectively. The addition of SIP processes also has a

great influence on the charging separation. On average, the mean positive and negative noninductive charging rates in the 4SIP experiment are 3.9 times and 46% greater than in the noSIP experiment, and the positive inductive charging rates can be enhanced by 1.2 times by implementing the four SIP processes. In addition, the aerosol concentration and the SIP processes can affect the area of the positive and negative charging. As $N_0$ increases from 400 $cm^{-3}$ to 4000 $cm^{-3}$, the areas of positive noninductive and inductive charging increase by 18% and 23%, respectively. With the addition of SIP processes, the area of positive noninductive and negative inductive rates decreases by about 3% and 23%, while the area of negative noninductive and positive inductive charging increases.

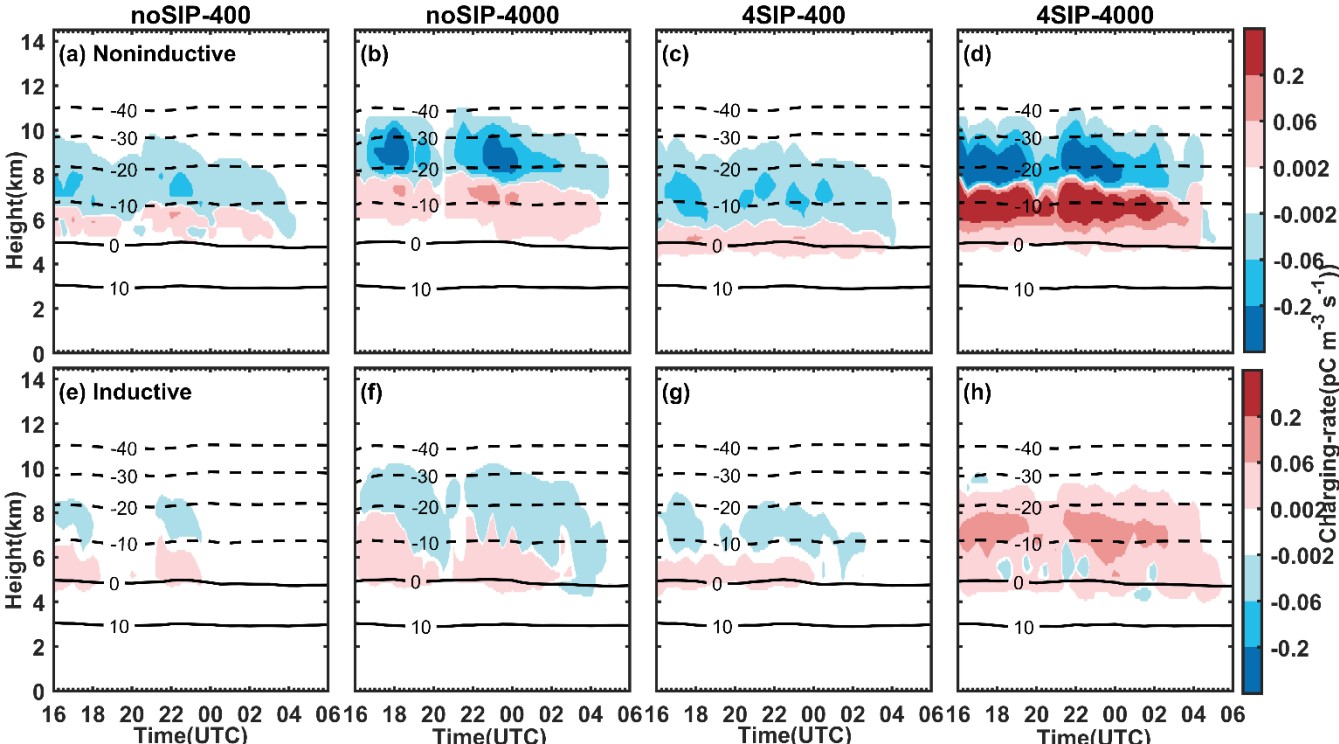

**Figure 11. Time-height evolution of the mean charging rate (unit is pC/m³·s) for four sensitivity experiments. (a-d) noninductive charging rate, (e-h) inductive charging rate. The first to fourth columns represent the results from noSIP-400, noSIP-4000, 4SIP-400, and 4SIP-4000 experiments, respectively.**

The vertical profiles of the mean noninductive charging rate and mean inductive charging rate are shown in Figure 12, including the sensitivity tests using individual SIP processes. In all the experiments, the noninductive charging rate shows an upper negative and lower positive pattern (Fig. 12a-f). The magnitude of the noninductive charging rate increases as the CCN concentration increases. The IC and SK process has a minor effect on the noninductive charging rate (Fig. 12c and e), as the IC process mainly affects the upper-level ice production, and the SK process is very weak in the present case. For $N_0$=400 $cm^{-3}$, the SD has the most significant impact on the noninductive charging rate. As $N_0$ increases to 4000 $cm^{-3}$, both the RS and SD

process can result in a significant increase in the noninductive charging rate, in particular, the lower positive noninductive charging rate (Fig. 11b and d). With all four SIP processes considered and $N_0$=4000 cm$^{-3}$, the low-level positive noninductive charging rate is the strongest among all the experiments (Fig. 12f). The magnitude of the inductive charging rate can be enhanced as the CCN concentration increases, but the structure remains similar (Fig. 12g). However, the RS and SD processes not only intensify the inductive charging rate, but also significantly modify its vertical structure (Fig. 12h and j), and these two SIP processes dominate the vertical profiles of inductive charging rate in the experiments with all four SIP processes turned on (Fig. 12l).

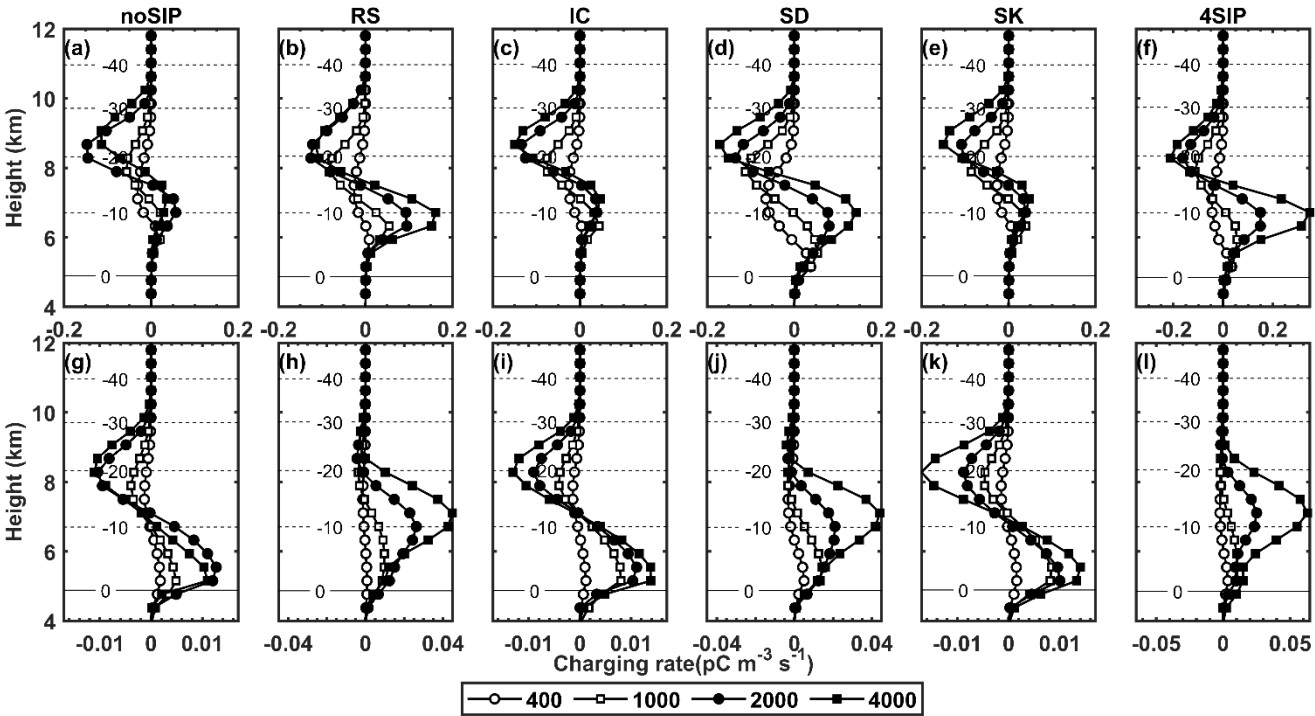

**Figure 12. The vertical profiles of the mean noninductive charging rate (upper panels) and mean inductive charging rate (lower panels). (a, g) noSIP, (b, h) RS only, (c, i) IC only, (d, j) SD only, (e, k) SK only and (f, l) 4SIP.**

According to the analysis in Section 3.2, both ice and graupel concentrations and mixing ratios are lower in noSIP-1000 than in noSIP-400, so it is expected that the collision rate between graupel and ice would be weaker when $N_0$=1000 cm$^{-3}$, implying a negative effect on the noninductive charging. However, the modelled noninductive charging rate is greater in noSIP-1000 than in noSIP-400 (Fig. 12). The reason is the noninductive charging rate is not only a function of graupel-ice collision kernel, but also a function of RAR, which is related to the LWC. Figure 13 shows the RAR in the noSIP experiment under different aerosol concentrations. It is seen that the RAR increases with increasing CCN concentration, indicating it is the enhanced LWC carried by droplets above the freezing level that dominates the aerosol impact on electrification. In addition, as the CCN concentration increases, RAR>0.1 g m$^{-2}$ s$^{-1}$, which is the threshold of noninductive charging (Saunders and Peck, 1998),

extends to colder temperature regions. This is why a negative noninductive charging rate above the -30°C isotherm is present when $N_0$=4000 cm$^{-3}$ (Fig. 12a).

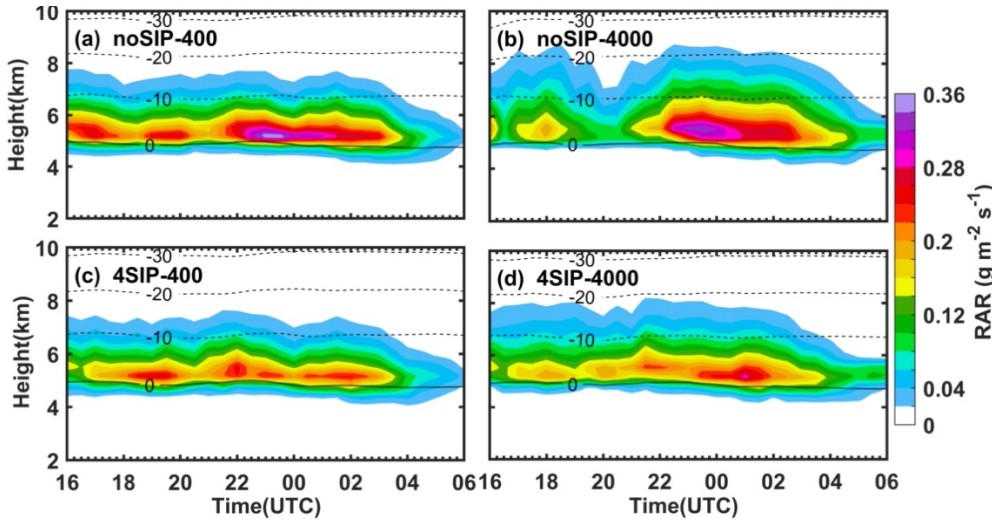


**Figure 13.Time-height evolution of the mean RAR (unit is gm$^{-2}$s$^{-1}$) for four sensitivity experiments. (a) noSIP-400, (b) noSIP-1000, (c) noSIP-2000, (d) noSIP-4000.**

The significant impacts of aerosol and SIP on charging rate results in different structures of charge density in different
experiments. Figure 14 shows the temporal evolution of charge density carried by graupel and ice/snow as well as the total charge density obtained from four sensitivity experiments. In the noSIP-400, noSIP-4000, and 4SIP-400 experiments, the graupel particles have negative charge densities on average (Fig. 14a-c), this is related to the sedimentation of graupel. As illustrated in Fig. 11, when $N_0$ is 400 cm$^{-3}$, the negative charge carried by the upper-level graupel is considerably stronger than the positive charge carried by the lower ones. However, charging separation takes place at a relatively small area at a given
time, thus, the negative charge carried by the falling graupel may exceed the positive charge transferred to the graupel at low levels. As $N_0$ increases from 400 cm$^{-3}$ to 4000 cm$^{-3}$, the total charge density exhibits intensification and the positive charge region are lifted up. The addition of SIP does not change the sign of charge density when the CCN concentration is low (Fig. 14c), but in the 4SIP-4000 experiment, a dipole structure is found (Fig. 14d). Due to the RS and SD processes, the positive charge transferred to the graupel through noninductive charging significantly increases below about -20°C, and exceeds the
magnitude of upper-level negative charging (Fig. 11d). Therefore, graupel particles are on average negatively charged in the region colder than -20°C and positively charged in the region warmer than -20°C (Fig. 14d). It is evident that increasing the aerosol concentration and the addition of SIP processes both can substantially modify the charge structure.

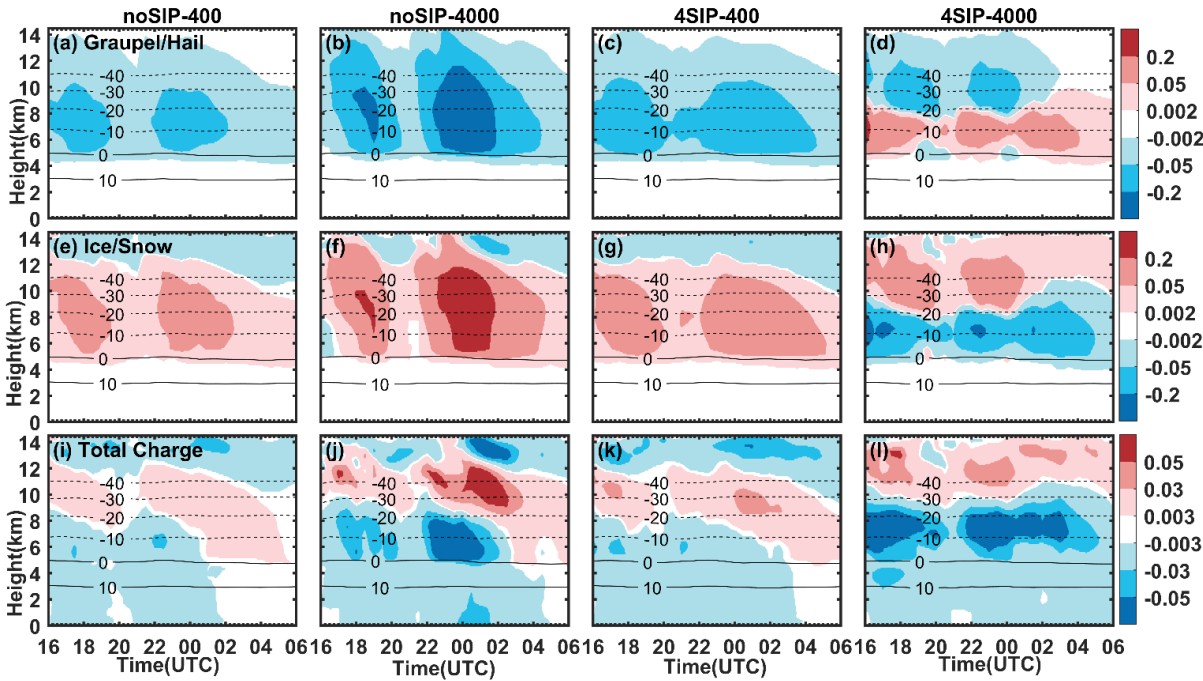

**Figure 14.** Time-height evolution of mean charge carried by graupel/hail (a-d) and ice/snow (e-h) particles as well as mean total space charge (i-l) (unit is $nC\ m^{-3}$) for four sensitivity experiments. The first to fourth columns represent the results from noSIP-400, noSIP-4000, 4SIP-400 and 4SIP-4000 experiment, respectively.

Since the RS and SD processes have greater impacts on cloud electrification in the present case, we investigate the average vertical profiles of charge density from the sensitivity tests with the RS and SD processes implemented (Figure 15). It is seen that as $N_0$ increases from 400 cm$^{-3}$ to 2000 cm$^{-3}$, the amount of charge carried by the graupel and ice/snow particles as well as the total charge increase markedly, while as $N_0$ increases from 2000 cm$^{-3}$ to 4000 cm$^{-3}$, the charge carried by the graupel and ice/snow particles decreases, this is consistent with the modelled noninductive charging rate shown in Fig. 12a. When $N_0$ is 400 or 1000 cm$^{-3}$, the addition of RS has little effect on the amount and distribution of space charges, which is probably because the cloud droplets are insufficient to induce a significant secondary ice production through the RS process. With the same $N_0$, the addition of SD enhances the amount of space charges, but it is not able to change the inverted tripole structure of the total charge density. It is the RS process in a polluted environment ($N_0 = 4000$ cm$^{-3}$) responsible for the generation of normal charge structure (Fig. 15n). Although we do not have observational data on the charge structure, the normal charge structure is more frequently observed in south China than inverted structure as shown by previous studies (Zhang et al., 1997). The combination of four different SIP processes can extend the positive charge region to higher levels.

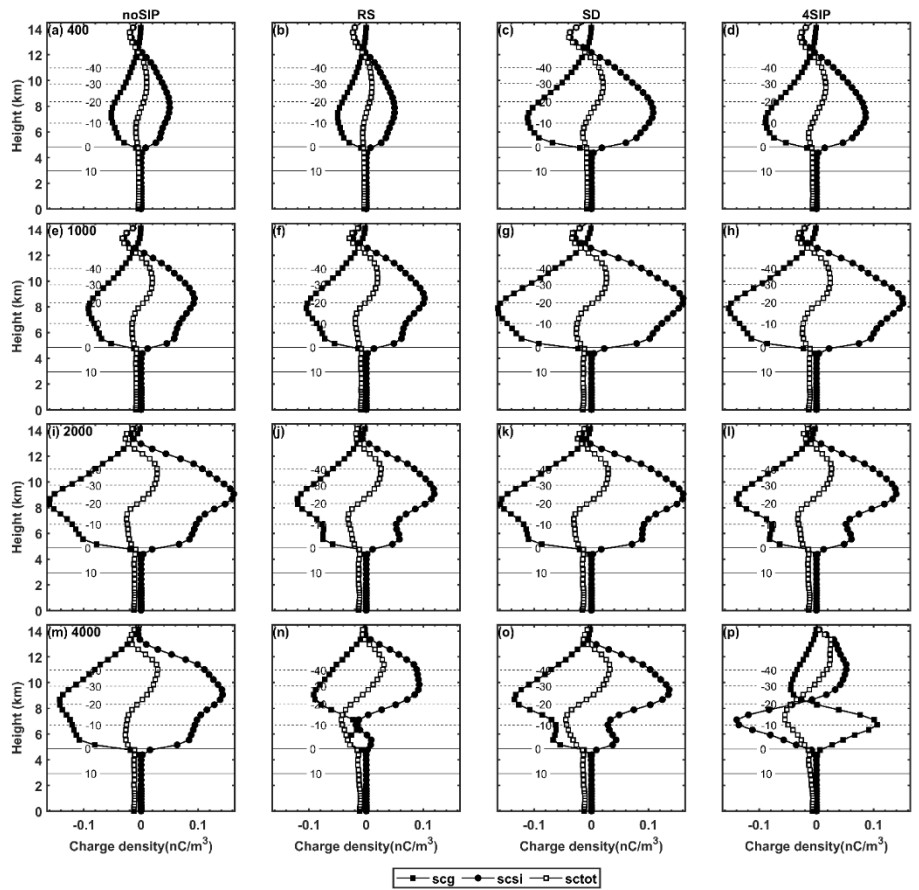


**Figure 15. The vertical profiles of mean charge carried by graupel/hail and ice/snow particles as well as mean total space charge for different sensitivity experiments. (a-d) $N_0$=400 cm$^{-3}$, (e-h) $N_0$=1000 cm$^{-3}$, (i-l) $N_0$=2000 cm$^{-3}$, (m-p) $N_0$=4000 cm$^{-3}$. The first to fourth columns represent the results from experiments with no SIP, RS only, SD only and all 4 SIP processes, respectively. Black box lines represent graupel charge density (scg), black dot lines represent ice/snow charge density (scsi) and white box lines represent total charge density (sctot).**

Modification of space charge density by aerosol and SIP processes would certainly influence the electric field. The time–height variations of the electric field obtained from four experiments are shown in Figure 16. The core of the maximum electric field is between 0℃ and -40 ℃, which coincides with the region where the charge is concentrated. Without any SIP process considered, the electric field after 22:00 UTC increases significantly as $N_0$ increases from 400 cm$^{-3}$ to 4000 cm$^{-3}$, which is consistent with the enhancement of total charge density shown in Fig. 14. In the presence of four SIP processes, high CCN concentration causes a remarkable increase in the electric filed between 0℃ and -20 ℃. It is clear the enhancement of the electric field is closely related to the enhancement of the total charge density. The comparison of electric fields obtained from

the four experiments reveals that both the addition of the SIP processes and the increase in aerosol concentration favor the enhancement of the electric field.

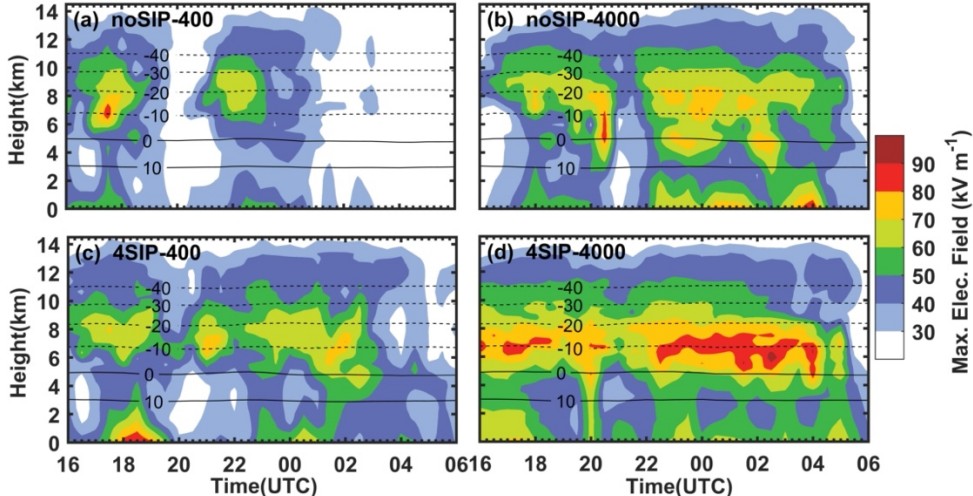

**Figure 16. Time-height evolution of the maximum electric field for the noSIP-400, noSIP-4000, 4SIP-400 and 4SIP-4000 experiment.**

To investigate the impacts of SIP processes and CCN concentration on lightning, the flash rate from both the noSIP and 4SIP experiments are shown in Fig. 17. The flash rate can be remarkably enhanced by increasing the CCN concentration, this is consistent with previous studies and again suggest the different aerosol concentrations is a key explanation for the ocean-land contrast of flash rate. Without SIP processes, the flash rate in the case with a CCN concentration of 4000 $cm^{-3}$ is 2-3 orders of magnitude greater than that with a CCN concentration of 400 $cm^{-3}$. The flash rate reaches a "plateau" beyond a CCN concentration of approximately 1000 $cm^{-3}$, as there is little difference between the noSIP-2000 and noSIP-4000 cases. The addition of SIP processes can also slightly enhance the flash rate. For the cases with a CCN concentration of 400 $cm^{-3}$ and 4000 $cm^{-3}$, the average flash rates between 16:00 May 29th and 04:00 May 30th increase by 66.8% and 44.3% due to the SIP processes, respectively. Therefore, the flash rate from the 4SIP-4000 experiment is the largest.

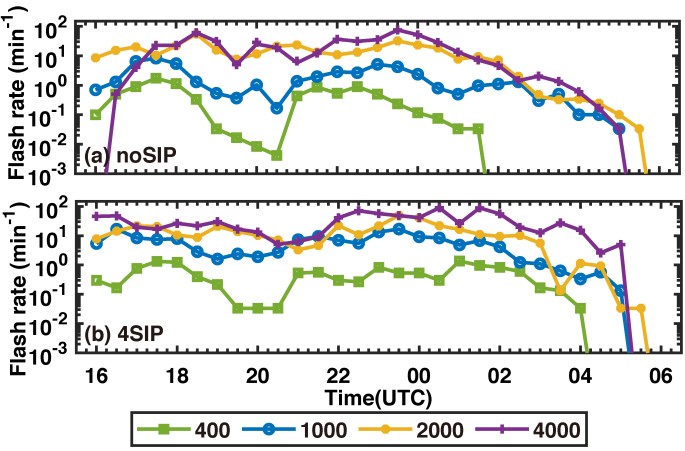

Figure 17. Time evolution of the flash rate for the noSIP experiment and 4SIP experiment.

## 4 Discussion and Conclusions

To investigate the effects of SIP processes and aerosols on cloud microphysics and electrification, the Fast-SBM microphysics scheme with the addition of parameterizations of SIP, noninductive and inductive charging processes, and a bulk discharge scheme are implemented to simulate a squall line occurred on May 29-30, 2022, in southeast China. The results are concluded

as follows:

(1)    The simulation results well reproduce the macro-morphology, the occurrence location, and the eastward tendency of this squall line. The addition of SIP processes and the concentration of aerosol particles have little effect on the macroscopic morphology of this squall line but have a significant effect on its intensity. The experiment with more realistic aerosol

concentration and all SIP processes gives the closest results to observation.

(2)    Cloud microphysics characteristics are sensitive to SIP processes and aerosol concentration. More cloud droplets can be lifted to higher cloud levels and the warm rain is suppressed as the CCN concentration increases. Fewer graupel particles can form due to fewer rain-size drops for freezing as the CCN concentration increases. Without any SIP process, the

convective updraft is inhibited as CCN concentration increases from 400 cm$^{-3}$ to 1000 cm$^{-3}$, leading to a lower total water mixing ratio and a lower ice concentration. However, the aerosol impact on ice microphysics exceeds its impact on dynamics as CCN concentration increases from 1000 cm$^{-3}$ to 4000 cm$^{-3}$. In a clean environment, the SD process is the most important SIP process to ice production between 0°C and -10°C, and the graupel concentration can be significantly enhanced. In a polluted environment, the RS process contributes the most to the graupel and ice production between 0°C

and -10°C. The ice-ice collisional break-up process contributes the most to ice production between -10°C and -20°C.

(3)   The noninductive charging rate is significantly greater than the inductive charging rate. The noninductive charging rates illustrate a distinct dipole structure having an upper negative region and a lower positive region. Note that the implementation of four SIP processes as well as the increase in aerosol concentration both cause an enhancement of the noninductive charging rate. However, aerosol and SIP processes have opposite impacts on the charging reversal: higher CCN concentration results in a colder reversal temperature, while SIP processes lower the reversal level. The IC and SK processes have a minor effect on the noninductive charging rate. In a clean environment, the SD has the most significant impact on the noninductive charging rate. In a polluted environment, both the RS and SD processes can result in a significant increase in the noninductive charging rate. The RS and SD processes not only intensify the inductive charging rate, but also significantly modify its vertical structure.

(4)   Increasing the CCN concentration and the addition of SIP processes both can modify the charge structure. With higher aerosol concentration, the total charge density exhibits intensification and the positive charge region are lofted to higher levels. When the CCN concentration is low (400 $cm^{-3}$), the addition of the SD process can enhance the charge density, but the change in charge structure is minor. When the CCN concentration is high (4000 $cm^{-3}$), the charge structure can be significantly modified, and the RS is the most important SIP process to induce the changes.

(5)   Both the addition of the SIP processes and the increase in aerosol concentration favor the enhancement of the electric field in the thunderstorm. The flash rate can be remarkably enhanced by increasing the CCN concentration, while the addition of SIP processes only slightly enhances the flash rate.

The results of this paper emphasize the necessity of adding SIP processes in the numerical model and the importance of aerosol concentration for numerical simulations. Studying the effects of SIP processes and aerosol concentrations on cloud microphysics and electrification simultaneously is a new thinking of thunderstorm simulation. In this study, aerosol concentrations from 400 to 4000 $cm^{-3}$ are considered, the results suggest generally a higher aerosol concentration leads to stronger charge separation, but the aerosol impact on cloud microphysics and electrification is not linear. This is also found in some previous studies, for example, Mansell and Ziegler (2013) tested 13 different aerosol concentrations from 50 to 8000 $cm^{-3}$ to investigate the effect of aerosols on storm electrification and precipitation. They found that the graupel concentration increases as the CCN concentration increases from 50 to 2000 $cm^{-3}$, but slowly decreases as the CCN concentration increases from 2000 $cm^{-3}$. Tan et al. (2015) designed simulation experiments with CCN concentrations from 50 to 10000 $cm^{-3}$. They found that more cloud droplets, graupel, and ice crystal production lead to a stronger charge separation as aerosol concentration increases from 50 to 1000 $cm^{-3}$. In contrast, as the aerosol concentration increases from 1000 to 3000 $cm^{-3}$, the mixing ratio of ice crystals decreases, the noninductive charging is weakened, while the inductive charging rate has no significant change.

The stronger charge separation induced by higher aerosol concentration may modify the structure of total charge density. For example, Shi et al. (2019) found that the charge structure at different convective intensities (by controlling the environmental humidity and temperature stratification at an initial time) became more complex as the aerosol increased. Sun et al., (2024) showed that compared to the low aerosol concentration case, a notable inverted dipole charge structure was simulated in the high aerosol environment. The modelled charge structure in different cases may be different, depending on multiple factors such as the thermodynamic properties and LWC (Phillips and Patade, 2022; Zhao et al., 2020), and is possibly related to the different parameterizations of electrification used in various studies (Phillips and Patade, 2022). Nevertheless, all these studies, including the present paper, demonstrate the flash rate can be enhanced by higher aerosol concentration, which is regarded as a key explanation for the higher flash rate over continents than over ocean.

Previous studies have pointed out that the SIP processes strongly affect cloud microphysics and electrification (Waman et al., 2022; Huang et al., 2022; Phillips and Patade, 2022). In this study, we further show that the RS process is the most important one in an environment with high aerosol concentration, and the SD process is more important when the aerosol concentration is low. This conclusion is consistent with previous studies which suggest the RS process can strongly affect the charge separation in continental thunderstorms (Huang et al. 2024; Yang et al. 2024), and the SD process may be a more efficient SIP mechanism in maritime convection, in which more supercooled rain drops are observed (Field et al., 2017). Phillips and Patade (2022) investigated a convective cloud with a cold base, they suggested the IC process is more active than the RS and SD process as the droplets are too small. In our case, the IC process is only efficient at temperatures colder than -10 °C in the mature stage. The sublimational breakup process has the least impact, which is also found in mature convective clouds simulated by Waman et al. (2022).

Regardless of the differences in various studies, it is commonly found that the aerosol concentration and SIP processes both have great impacts on cloud microphysics and electrification. An increase in aerosol concentration leads to a nonlinear enhancement of the charging rate. The RS process is the vital SIP process in a polluted environment, and the SD process is more important in a clean environment. Therefore, accurate representations of the various SIP processes under different aerosol conditions are important for the model simulation of lightning activities.

**Data availability**

The WRF model is available at https://www2.mmm.ucar.edu/wrf/users/download/get_source.html (NCAR MMM, 2023). The modified Fast-SBM scheme, the reanalysis data used to drive the WRF model, the observed radar reflectivity, and FY4A data are available at https://doi.org/10.5281/zenodo.13910807 (Yang, 2024).

## Author contributions

SH and JY implemented the parametrizations of SIP and electrification in WRF, and designed the numerical experiments. SH, JY, JL, and QC performed the analysis and prepared the manuscript. QZ and FG contribute to the model evaluation. QZ and FG provided input on the method and analysis. All authors provided significant feedback on the manuscript.

## Competing interests

The contact author has declared that none of the authors has any competing interests.

## Acknowledgments

This work was supported by the National Natural Science Foundation of China (42475090, 41905124, 41975003), the Natural Science Foundation of Jiangsu Province, China (BK20190778), and the CMA Key Innovation Team Support Project (CMA2022ZD10). We acknowledge the High Performance Computing Center of Nanjing University of Information Science & Technology for their support of this work. We appreciate the editor and reviewers for their insightful comments and suggestions.

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
