# Peer review of "Impact of secondary ice production on thunderstorm electrification under different aerosol conditions"

_EGUsphere, 2024_

## Author Comment (AC3)

Reviewer's comments are in black, and responses are in blue.

**General comments:**

Huang et al. present a detailed investigation into the role of secondary ice production (SIP) processes and aerosols in cloud electrification within thunderstorms. Utilizing the Weather Research and Forecasting (WRF) model with a spectral bin microphysics (SBM) scheme, this study examines how different SIP processes interact with varying cloud condensation nuclei (CCN) concentrations to influence cloud microphysics and charge separation. This topic is both timely and relevant, exploring key uncertainties in our understanding of cloud microphysics and electrification mechanisms under different aerosol conditions. Nevertheless, the manuscript requires substantial revisions to improve clarity and strengthen the validity of conclusions. The following specific comments and technical corrections should be addressed before the paper is considered for publication:

Reply: We appreciate your insightful comments. The paper has been revised accordingly and has been improved a lot. Please see our responses below.

**Specific comments:**

1. **In the abstract:** (i) Please be more precise when referring to terms like "clean" and "polluted" environments, as well as "relatively warm/cold temperatures," both in the abstract and throughout the manuscript. Explicitly define these terms or specify the aerosol conditions and temperature ranges you are referring to each time to avoid ambiguity. (ii) It would be beneficial to mention some implications of your findings within the abstract. For example, you could explain the importance of correctly capturing the charging reversal at different temperatures due to variations in aerosol concentrations or the inclusion of SIP. This would help readers understand the relevance and impact of your results. (iii) Finally, please define what is meant by the term "normal charge structure" (Line 29).

Reply: Thank you for your comment. (i) The aerosol conditions and temperature ranges have been clearly described to avoid ambiguity. (ii) The charging reversal at different temperatures is important because it affects the polarity of the charge acquired by the ice-phase particles during charge separation, which in turn affects the charge structure and the flash rate in the cloud. According to the analysis in this paper, it is known that the reversal temperature is affected by both aerosol concentration and SIP processes. It

implies the importance of aerosol and SIP processes for the charge structure and discharge process in clouds. (iii) The term "normal charge structure" is defined as "with an upper positive charge region and a lower negative region" in the abstract.

2. **Lines 41-52:** Here you mention the aerosol indirect effects in liquid clouds (cloud albedo and lifetime effect). Consider also discussing the indirect aerosol effects in mixed-phase clouds (MPCs; e.g., glaciation, riming and thermodynamic indirect effect in convective clouds). This is particularly relevant to your study, and these indirect effects could help explain some of the results you present (e.g., Lines 241-247 or 296-298).

Reply: Thank you for your comment. The indirect aerosol effects in mixed-phase clouds is discussed as follows: "*Aerosol also has significant impacts on the microphysics in mixed-phase region. According to previous studies, the increase in aerosol concentration reduces cloud droplet size, thus more small droplets can be lofted to mixed-phase region, which may enhance the freezing of cloud droplets and facilitate cold rain process (Rosenfeld and Woodley, 2000; Rosenfeld et al., 2008; Hoose et al., 2010; Sherwood, 2002; Jiang et al., 2008). The increase in CCN condensation may also intensify the hail growth through riming in convective clouds (Chen e al., 2010), and the enhanced ice growth rate may produce more latent heat, which in turn invigorate the convections (Khain et al., 2005; Wang, 2005).*".

References:

Cheng, C.-T., Wang, W.-C., and Chen, J.-P.: Simulation of the effects of increasing cloud condensation nuclei on mixed-phase clouds and precipitation of a front system, Atmospheric Research, 96, 461–476, https://doi.org/10.1016/j.atmosres.2010.02.005, 2010.

Hoose, C., Kristjánsson, J. E., and Burrows, S. M.: How important is biological ice nucleation in clouds on a global scale?, Environ. Res. Lett., 5, 024009, https://doi.org/10.1088/1748-9326/5/2/024009, 2010.

Jiang, J. H., Su, H., Schoeberl, M. R., Massie, S. T., Colarco, P., Platnick, S., and Livesey, N. J.: Clean and polluted clouds: Relationships among pollution, ice clouds, and precipitation in South America, Geophysical Research Letters, 35, 2008GL034631, https://doi.org/10.1029/2008GL034631, 2008.

Khain, A., Rosenfeld, D., and Pokrovsky, A.: Aerosol impact on the dynamics and microphysics of deep convective clouds, Q. J. R. Meteorol. Soc., 131, 2639–2663, https://doi.org/10.1256/qj.04.62, 2005.

Sherwood, S. C.: Aerosols and Ice Particle Size in Tropical Cumulonimbus, J. Climate, 15, 1051–1063, https://doi.org/10.1175/1520-0442(2002)015<1051:AAIPSI>2.0.CO;2, 2002.

Wang, C.: A modeling study of the response of tropical deep convection to the increase of cloud condensation nuclei concentration: 1. Dynamics and microphysics, J. Geophys. Res., 110, 2004JD005720, https://doi.org/10.1029/2004JD005720, 2005.

3. **Lines 54-63:** Before discussing the results from various model simulations, it would be beneficial to provide a brief introduction to SIP. Additionally, introducing Primary Ice Production (PIP) as a prerequisite for SIP would clarify the mechanisms by which an increased presence of cloud droplets promotes ice formation in MPCs.

Reply: Thank you for your valuable comment. The relative contents are added as follows: "*The production of ice crystals in thunderstorm clouds is complicated, in mixed-phase region, the heterogeneous nucleation, which depends on ice nucleating particles (INP), provides the primary ice. However, it is found that the primary ice production (PIP) is insufficient to explain the observed high ice concentration in convective clouds (Hobbs, 1969; Hobbs and Rangno, 1985; Koenig, 1963). Secondary ice production (SIP) is recognized as a major contributor to the fast ice generation at temperatures warmer than the homogeneous freezing temperature (Hallett and Mossop, 1974; Mossop and Hallett, 1974; Santachiara et al., 2014; Korolev 2020). There are several different mechanisms of SIP proposed by previous studies, such as ice splintering during during riming (Hallet and Mossop, 1974), shattering of drops during freezing (Gagin, 1972; Pruppacher and Schlamp,1975; Kolomeychuk et al., 1975), ice-ice collisional breakup (Schwarzenboeck et al., 2009), and ice sublimation breakup (Bacon et al., 1998; Dong et al., 1994; Oraltay and Hallett, 1989). The ice concentration produce by SIP processes are orders of magnitude than PIP, but PIP is a prerequisite for SIP, and changes in PIP (e.g., induced by increasing aerosol concentration) can affect the SIP processes (Sullivan et al., 2018).*


Reply: Thank you for your valuable comment. Despite the deviation between the observed and simulated values, the simulation results well reproduce the macro-morphology, the occurrence location, and the eastward tendency of this squall line. The observed result is the same as Huang et al. (2024). For the southeastern coast of China, both FNL (Guo et al., 2017; Qu et al., 2017) and ERA5 (Gong et al., 2024; Wang et al., 2023) data are widely used in numerical simulations. Based on your suggestion, we compare the simulated radar reflectivity of these two data. Figure R1 shows that simulated radar reflectivity from 4SIP-4000 experiment with ERA5 reanalysis data. It is noted that simulation results obtained for FNL and ERA5 data are similar. Therefore, FNL data is kept for use in this paper.

[Figure]

Figure R1. (a) The simulated radar reflectivity from 4SIP-4000 experiment with ERA5 reanalysis data. (b) The observed radar reflectivity.

Table R1. The difference in mean rain mixing ratio and number concentration between noSIP experiment and SD experiment (SD-noSIP) at 0°C to -10°C.

| $N_0$ | Mixing ratio | Number concentration |
|---|---|---|
| 400 | -60.57% | -43.57% |
| 1000 | -75.34% | -62.31% |
| 2000 | -99.42% | -75.87% |
| 4000 | -59.88% | -37.99% |

14. **Figure 6:** Confirm whether the mean vertical profiles in this figure correspond to the same period shown in Figure 5. Additionally, have you tried to use logarithmic scale for the horizontal axes showing the number concentration of particles in this figure?

Reply: Thank you for your comment. These two figures in the revised manuscript share the same period. Figure R2 uses the logarithmic scale on the horizontal axes. In our opinion, the linear scale in the original figure better reveals the difference among different experiments. For example, the difference of graupel number concentration among the four experiments is more pronounced using liner scale axis.

[Figure]

Figure R2. The vertical profiles of mean concentration for (a) cloud drop, (b) rain, (c) graupel/hail and (d) ice/snow from noSIP-400, noSIP-4000, 4SIP-400 and 4SIP-4000 experiment. The horizontal axes are logarithmic scale.

15. **Figure 7:** This figure appears overly complex, potentially obscuring key insights from your sensitivity experiments. Please consider reducing the number of sensitivities discussed in the main paper or discussing any plateau or asymptotic behavior as CCN concentrations increase. A logarithmic vertical axis might also be beneficial in this figure. Is there an issue with the y-axis in Figure 7b?

Reply: Thank you for your comment. According to your suggestion, the figure is revised and the y-axis in (b) is modified. Now only noSIP and 4SIP results are shown in the figure (Fig. R3). In fact, we only discussed noSIP and 4SIP results in the original paper, so the text remains similar. We tried using logarithmic vertical axis but in our opinion the linear scale in the original figure better reveals the difference among different experiments.

[Figure]

Figure R3. The time-domain-averaged mixing ratios (a, c, e, g) and number concentrations (b, d, f, h) of (a, b) cloud drop, (c, d) rain, (e, f) graupel/hail and (g, h) ice/snow from 8 sensitivity experiments.

[Figure]

Figure R4. The same as Fig. R3, but the vertical axis are logarithmic.

16. **Line 285-286:** Have you examined how SIP rates vary with changes in CCN concentrations?

Reply: Thank you for your valuable comment. Yes, the secondary ice production rate of four SIP processes has been discussed in Fig. 9 (Fig. R5 here) in the original manuscript.

[Figure]

Figure R5. Time-height revolution of secondary ice production rate of four mechanisms of SIP. (a-d) rime-splintering process (RS); (e-h) ice-ice collisional breakup (IC); (i-l) freezing drop shattering (SD) and (m-p) sublimational breakup (SK). The first to fourth column indicate $N_0$=400, 1000, 2000, 4000 cm$^{-3}$, respectively.

17. **Line 313 and Figure 9:** Please elaborate on why IC is most significant within the -10°C to -20°C temperature range. The enhanced efficiency of IC within the so-called dendritic growth layer is discussed in von Terzi et al. (2022) and Georgakaki et al. (2024). Did you implement both IC formulations from Table 1 of Phillips et al. (2017) for dendritic and spatial planar ice crystal habits? If so, was the selection of ice habit based on temperature? This should be further clarified in Section 2.2.1, as it might help explain the results discussed here. Also, please consider using a logarithmic scale for the colorbar in this figure.

Reply: Thank you for your comment. As shown in Von Terzi et al. (2022) and Georgakaki et al. (2024), collision of dendritic ice is the reason for enhanced IC process between -10°C and -20°C. The greater ice number concnetration between -10°C and -20°C favors the ice collision, while smaller size of ice above -20°C isotherm is unfavorable for the collision between the ice.

Yes, we do implement both IC formulations from Table 1 of Phillips et al. (2017) for dendritic and spatial planar ice crystal habits. Since the fast-SBM does not separate different ice habits, the selection of ice habit is simply based on temperature. Ice is assumed to be dendritic between -12°C and -17°C and is spatial planar ice between -40°C and -17°C as well as between -9°C and -12°C.

Figure R6 is the same as Fig. 9 in the original manuscript, but the colorbars have logarithmic scales. In our opinion, the main difference of production rates is clearer with liner colobar axis than logarithmic colobar.

[Figure]

Figure R6. Time-height evolution of secondary ice production rate of four mechanisms of SIP. (a-d) rime-splintering process (RS); (e-h) ice-ice collisional breakup (IC); (i-l) freezing drop shattering (SD) and (m-p) sublimational breakup (SK). The first to fourth column indicate $N_0$=400, 1000, 2000, 4000 cm$^{-3}$, respectively. The colorbars have logarithmic scales.


Reply: Thank you for your comment. Since the temperature contours in the time-height plot are obtained by taking the mean value, there is inevitably a bias. In order to better show the distribution of production rates, the cross-section plot is shown. As can be clearly seen in Figure R7, the generation rate is restricted to between -3°C and -8°C.

[Figure]

Figure R7. The cross-section of production rate of rime splintering process. (a) $N_0$=400 cm$^{-3}$; (b) $N_0$=1000 cm$^{-3}$; (c) $N_0$=2000 cm$^{-3}$; (d) $N_0$=4000 cm$^{-3}$;

19. **Section 3.3:** Is there any way to evaluate the modeling results and conclusions from this section? In your previous work (Yang et al., 2024), you included flashing rates observations. Were such observations available for the squall line discussed in this study?

Reply: Thank you for your comment. We are sorry for not having observed flash rate data using ground-based stations. However, the comparison of observed (from FY-4A satellite) and simulated lightning locations is shown in Figure 5 (Fig. R8 here) in the revised manuscript. The describes is as follows: "*In this paper, the observed lightning data are collected by the Lightning Mapping Imager (LMI), which is mounted on the second-generation Chinese geostationary meteorological satellite FY-4A to continuously detect lightning in China and its neighbouring areas. Additionally, the brightness temperature captured by FY-2H satellite is used to delineate the area of the deep convective cloud. Figure 5 shows the location of the lightning events observed by LMI and simulated in 4SIP-4000 experiment. It is evident that the simulated and observed lightning locations are in good agreement with each other and are both in the low brightness temperature region, which characterizes the presence of strong convection.*"

[Figure]

Figure R8. The simulated and observed lightning locations (sign "+"). The shaded field indicates brightness temperature.

20. **Lines 435-437:** Could you clarify the need to alter the "inverted tripole structure" of the total charge density? What is meant by "normal charge structure," and to which subplot in Figure 15 does this refer?

Reply: Thank you for your valuable comment. The normal charge structure has an upper positive charge region and a lower negative region and is shown in Fig. 16n and d in the original manuscript. Although we do not have observational data on the charge structure, the normal charge structure is more frequently observed in south China than inverted structure as shown by previous studies (Zhang et al., 1997).

Reply: Thank you for your comment. The "concertation" has been changed to "concentration" in the revised manuscript.

---

## Author Response (AR1)

**Response to reviewers**

Dear Editor and Reviewers,

We would like to thank the editor and reviewers for your professional comments and suggestions, which are very helpful in improving our manuscript. We have carefully considered the critical comments and thoughtful suggestions by reviewers, responded to these suggestions in a point-by-point manner, and revised the manuscript accordingly. In the Tracked-Changed Manuscript, changes are marked using the Review Mode of Microsoft WORD. Our point-by-point response to the comments raised by reviewers is shown as follows. The comments are in black, and our responses are given directly afterward in blue.

**Reply to the comments of Reviewer 1**

**General comments:**

Huang et al. investigated the role of secondary ice production on thunderstorm electrification under various aerosol conditions using numerical simulations. Overall, the research topic is interesting and valuable for the scientific community. However, in its current form, the manuscript needs major changes before being considered for further revision or final acceptance. I have enlisted my specific and minor comments below.

Reply: We appreciate your insightful comments. The paper has been revised accordingly and has been improved a lot. Please see our responses below.

**Specific comments:**

1. Model validation is done based only on radar reflectivity. Since the main objective of the paper is thunderstorm electrification it is important to validate additional variables from model simulations with observations such as flash rate, rain rate, ice number concentration, ice water path, etc. It will be interesting to see how changes in aerosol affect the flash rate in the presence of SIP mechanisms.

Reply: Thank you for your comment. Unfortunately, we don't have in-situ measurements of cloud microphysics such as ice number concentration, ice water path, etc. CloudSat data is not available for this case to retrieve the ice microphysics. The locations of lightning can be obtained from the FY4A stationary satellite measurement, we add the comparison between the modelled and observed lightning location in the revised paper (Fig. R1). The limitation of FY4A is its detection efficiency is relatively low during the daytime because of the reflectance of sunlight (Sun et al., 2022), and the flash rate cannot be obtained from FY4A. Statistically, the locations of the observed and modelled lightning are consistent.

According to your Comment 2, the 4SIP-4000 experiment has been set as a control experiment and the following analysis has been added to the revised manuscript: "*In this paper, the observed lightning data are collected by the Lightning Mapping Imager (LMI), which is mounted on the second-generation Chinese geostationary meteorological satellite FY-4A to continuously detect lightning activity in China and its neighboring areas. Additionally, the brightness temperature captured by FY4A satellite is used to delineate the area of the deep convective cloud. Figure R1 shows the location of the lightning events observed by LMI and simulated in 4SIP-4000 experiment. It is seen that the simulated and observed lightning locations are in good agreement with each other and are both in the low brightness temperature region, which characterizes the presence of strong convection.*"

[Figure]

Figure R1. The simulated and observed lightning locations (sign "+"). The shaded field indicates brightness temperature.

To investigate the impact of aerosol on the flash rate in the presence of SIP, we plot Fig. R2, which shows the flash rates modelled in the 4SIP experiments with different aerosol concentrations. The results indicate that a higher aerosol concentration can lead to a greater flash rate, which is consistent with previous studies (e.g., Liu et al., 2020; Shukla et al., 2022; Wang et al., 2011)

[Figure]

Figure R2. Flash rate modelled by the 4SIP-400, 4SIP-1000, 4SIP-2000, and 4SIP-4000 experiments.

References:
Liu, Y., Guha, A., Said, R., Williams, E., Lapierre, J., Stock, M., and Heckman, S.: Aerosol Effects on Lightning Characteristics: A Comparison of Polluted and Clean Regimes, Geophysical Research Letters, 47, e2019GL086825, https://doi.org/10.1029/2019GL086825, 2020.

Shukla, B. P., John, J., Padmakumari, B., Das, D., Thirugnanasambantham, D., and Gairola, R. M.: Did dust intrusion and lofting escalate the catastrophic widespread lightning on 16th April 2019, India?, Atmospheric Research, 266, 105933, https://doi.org/10.1016/j.atmosres.2021.105933, 2022.

Wang, Y., Wan, Q., Meng, W., Liao, F., Tan, H., and Zhang, R.: Long-term impacts of aerosols on precipitation and lightning over the Pearl River Delta megacity area in China, Atmos. Chem. Phys., 11, 12421–12436, https://doi.org/10.5194/acp-11-12421-2011, 2011.

Sun, H., Yang, J., Zhang, Q., Song, L., Gao, H., Jing, X., Lin, G., and Yang, K.: Effects of Day/Night Factor on the Detection Performance of FY4A Lightning Mapping Imager in Hainan, China, Remote Sensing, 13, 2200, https://doi.org/10.3390/rs13112200, 2021.

2.   Also, it is not a good idea to compare perturbed simulations with observations and then pick the simulations that are in better agreement with observations. Authors should set a control simulation that is more realistic considering observed conditions of aerosol/CCN, including all SIP (like in real clouds). Then compare only control simulation with observations as a part of validation. This will increase the readability of the manuscript. Also, it is important to quantify the validation results. e.g. if there is bias in observations and simulations it should be mentioned in % or actual bias.

Reply: Thank you for your comment. The aerosol concentration in mainland China is typically high, we do not have direct measurements of aerosol concentration for this case, so we refer to previous studies in the same region. Qu et al. (2017) suggested using an $N_0$ of 4000 cm$^{-3}$ for southeast China. Therefore, we use the 4SIP-4000 experiment as the control experiment. In the revised paper, we now compare the control simulation with observation (Fig. R3), and the related text has been modified accordingly. The bias between observations and simulations has been quantified as follows: *"Comparison of the simulated results and observation reveals that the mean absolute errors of reflectivity at 00:00, 02:00 and 04:00 are 12 dBZ, 11 dBZ and 12 dBZ, respectively. The area where the radar reflectivity difference is less than 15 dBZ are 67%, 71%, and 67% at 00:00, 02:00 and 04:00, respectively."*

[Figure]

Figure R3. The (d-f) observed and (a-c) simulated radar reflectivity at 00:00, 02:00, 04:00 on 30th May. The simulated results are from 4SIP-4000 experiment.

References:

Qu, Y., Chen, B., Ming, J., Lynn, B. H., and Yang, M.-J.: Aerosol Impacts on the Structure, Intensity, and Precipitation of the Landfalling Typhoon Saomai (2006): Aerosol Impacts on Typhoon Saomai (2006), J. Geophys. Res. Atmos., 122, 11,825-11,842, https://doi.org/10.1002/2017JD027151, 2017.

3.    The Author needs to justify the need for 26 simulations in the current study. Many of those simulations are not important. In their previous studies, the role of all major SIP mechanisms as well as individual SIP mechanisms on deep cloud properties including electrification has already been investigated (e.g. Yang et al. 2024). The focus of the current study should be only on how different aerosol conditions alter cloud electrification together with SIP. In my opinion, there is no need for any simulation without individual SIP mechanisms. Only the changes in the rate of SIP production can be shown under various aerosol conditions (with and without all SIP) and their subsequent effect on cloud electrification should be discussed. In the current format, there is a lot of confusion in following the results.

Reply: Thank you for your comment. We agree, the focus of this paper is on the differences in cloud microphysics and cloud electrification under different aerosol conditions with and without SIP processes. In most of the figures, only noSIP and allSIP experiments are used. Other sensitivity experiments with individual SIP processes are

only used for clarification. Based on your suggestion, we have modified Table 1 and related text in the revised manuscript to make it easier for readers to understand.

4.  Line 141: Please describe briefly how the sublimation breakup was implemented in the model. Does the model have separate species of dendritic and non-dendritic crystals? If the implementation in your model has been described in the previous study give proper references.

Reply: Thank you for your comment. The Fast-SBM scheme in WRF does not distinguish the shape of the ice crystals (Khain et al., 2009). It treats solid particles as ice/snow or graupel/hail. Therefore, in this study, the sublimational breakup process is applied to ice/snow crystals. This parameterization has been used in previous studies (e.g., Yang et al., 2024; Deshmukh et al., 2022; Waman et al., 2022)

Reference:
Deshmukh, A., Phillips, V. T. J., Bansemer, A., Patade, S., and Waman, D.: New Empirical Formulation for the Sublimational Breakup of Graupel and Dendritic Snow, Journal of the Atmospheric Sciences, 79, 317–336, https://doi.org/10.1175/JAS-D-20-0275.1, 2022.

Khain, Leung, L. R., Lynn, B., and Ghan, S.: Effects of aerosols on the dynamics and microphysics of squall lines simulated by spectral bin and bulk parameterization schemes, J. Geophys. Res., 114, D22203, https://doi.org/10.1029/2009JD011902, 2009.

Waman, D., Patade, S., Jadav, A., Deshmukh, A., Gupta, A. K., Phillips, V. T. J., Bansemer, A., and DeMott, P. J.: Dependencies of Four Mechanisms of Secondary Ice Production on Cloud-Top Temperature in a Continental Convective Storm, Journal of the Atmospheric Sciences, 79, 3375–3404, https://doi.org/10.1175/JAS-D-21-0278.1, 2022.

Yang, J., Huang, S., Yang, T., Zhang, Q., Deng, Y., and Liu, Y.: Impact of ice multiplication on the cloud electrification of a cold-season thunderstorm: a numerical case study, Atmos. Chem. Phys., 24, 5989–6010, https://doi.org/10.5194/acp-24-5989-2024, 2024.

5.  The reasons behind the noninductive dipole structure having an upper negative region and lower positive region are not discussed properly/or are difficult to follow. Are there previous studies showing this kind of charge structure? If there is an increase/decrease in electrification due to SIP/aerosol should be quantified.

Reply: Thank you for your comment. In the SP98 noninductive charging scheme, the riming accretion rate (RAR) controls the charge sign during charge separation, and the

noninductive charging occurs where RAR is greater than 0.1 gm$^{-2}$s$^{-1}$ (Mansell et al., 2005; Saunders and Peck, 1998). The negative region denotes that an RAR is smaller than the critical RAR and the positive region denotes an RAR greater than the critical RAR. The critical RAR is a function of temperature. The RAR is the effective water content multiplied by the graupel fall velocity (Mansell et al., 2005). The relatively low liquid water content and the small size of graupel particles in the upper region lead to a smaller RAR, which is more likely to be less than the critical RAR and result in a negative noninductive charging rate. This is also found in previous studies, for example, Fierro and Mansell (2017) simulated idealized tropical cyclones and conducted sensitive experiments to investigate the impact of the wind shear and sea surface temperature on cloud microphysics and electrification. They found that positive charging occurs between 6.5 and 8 km with negative charging above that over a deeper layer, between 8 and 11 km. This discussion is added in the revised paper in the description of Fig. 11.

According to your comment, the bias in electrification due to SIP/aerosol has been quantified as follows: "*As $N_0$ increases from 400 cm$^{-3}$ to 4000 cm$^{-3}$, the noninductive charging strengthens, with the mean positive noninductive rate and negative rate increasing by 4.9 and 3.24 times, respectively. The inductive charging rate increases too, with positive and negative rates increasing by a factor of 8.7 and 4.9, respectively. The addition of SIP processes also has a great influence on the charging separation. On average, the rime splintering and freezing droplet shattering process has greater impacts than the other two. The mean positive and negative noninductive charging rates in the 4SIP experiment are 3.9 times and 46% greater than in the noSIP experiment, and the positive inductive charging rates can be enhanced by 1.2 times by implementing the four SIP processes. However, the negative inductive charging rate decreases 56.83% in 4SIP experiment than in noSIP experiment. In addition, the aerosol concentration and the SIP processes can affect the area of the positive and negative charging. As $N_0$ increases from 400 cm$^{-3}$ to 4000 cm$^{-3}$, the areas of positive noninductive and inductive charging increase by 18% and 23%, respectively. With the addition of SIP processes, the area of positive noninductive and negative inductive rates decreases by about 3% and 23%, while the area of negative noninductive and positive inductive charging increases.*"

[revised manuscript text omitted]

**Minor comments:**

1.  Line 17: delete up

Reply: Thank you for your comment. The "up" has been deleted in the revised manuscript.

2.  Line 82: Mention WRF model version

Reply: Thank you for your comment. The WRF model version is v3.9.1 and has been added to the revised manuscript.

3.  Line 97: Occurred in

Reply: Thank you for your comment. The "occurred at" has been changed as "occurred in" in the revised manuscript.

4.  Line 133: change depended to depending, filed to field

Reply: Thank you for your comment. These two have been revised in the manuscript.

5.  Line 134: change could to can

Reply: Thank you for your comment. The "could" has been changed to "can" in the revised manuscript.

6.  Line 148: SP98 should be defined here

Reply: Thank you for your comment. The SP98 is the name of the noninductive charging scheme proposed by Saunders and Peck (1998) and its definition is added in the revised paper.

7.  Line 152: signs

Reply: Thank you for your comment. The "sign" has been changed to "signs" in the revised manuscript.

8.  Line 185: Describe boundary conditions in this section

Reply: Thank you for your comment. The FNL reanalysis data with 0.25° × 0.25° resolution are used to provide the boundary conditions.

9.  Line 198: What was the slope value in the Twomey equation

Reply: Thank you for your comment. The slope value in the Twomey equation is 0.4.

10. Line 225: change that are to that is

Reply: Thank you for your comment. The "that are" has been changed to "that is" in the revised manuscript.

11. Line 227: mention bias in %

Reply: Thank you for your comment. The bias has been quantitatively described there.

12. Line 228: delete of

Reply: Thank you for your comment. The "of" has been deleted in the revised

manuscript.

13. Line 230: microphysics and electrification processes

Reply: Thank you for your comment. The "process" has been changed to "processes" in the revised manuscript.

14. Line 287: change than that in to than in

Reply: Thank you for your comment. The "that" has been deleted in the revised manuscript.

15. Figure 15: define legends scg, scsi etc

Reply: Thank you for your comment. The "scg", "scsi" and "sctot" have been defined in the revised manuscript.

**Reply to the comments of Reviewer 2**

**General comments:**

In fact, I also reviewed another paper by the author (Yang et al. 2024; https://doi.org/10.5194/acp-24-5989-2024), which can be considered a sister paper in some respects. In comparison to these two papers, the first one (Yang et al. 2024) seems more innovative because it appears to have established a new model. This paper only conducted sensitivity experiments with different aerosol concentrations using the model from the previous paper, which provided useful references but seemed to have lessened in terms of innovation.

Reply: We appreciate your comment. We agree that this study is based on the model in previous studies, although the technique is not new, the question we address is important and not well understood. To our knowledge, no study has investigated the role of different SIP processes in cloud electrification under different aerosol conditions. Our results show the RS process is the most important SIP process in a polluted environment, while the SD process is more important in a clean environment. We believe this can be a useful reference for future studies.

**Specific comments:**

1. First, we acknowledge that different secondary ice production (SIP) processes or different aerosol concentrations play an important role in cloud electrification, which has been previously studied in many related studies (Mansell and Ziegler 2013; Tan et al. 2015; Phillips and Patade 2022; Yang et al. 2024). At this stage, it seems less urgent to continue discussing the impact of different SIP processes on electrification under different aerosol conditions. Because cloud microphysical characteristics change successively after the SIP process and aerosol concentration changes, the part related to electricity will also naturally change. When the charging rate changes, the evolution of the charge structure is only a more superficial feature. We know that there are many experiments or hypotheses related to the SIP processes, but which process of SIP actually plays a role in clouds and whether they all play a role at the same time is still a question worth exploring. Therefore, we are still unsure whether 4SIP will play a role at the same time.

Reply: Thank you for your comment. We acknowledge that previous studies have revealed the effects of aerosol concentration or SIP processes, but to our knowledge, no study has explored in detail the role of different SIP processes under different aerosol conditions. Mansell and Ziegler (2013) tested 13 different aerosol concentrations but only considered the rime-splintering process. Tan et al. (2015) investigated the effect of different aerosol concentrations on electrification but not in-depth on the SIP processes. Phillips and Patade (2022) investigated the impact of aerosol concentration and three SIP processes, respectively, but they did not report which SIP dominates under

different aerosol conditions. These studies are good references, but we think this is worth conducting a study focusing on the role of different SIPs under various aerosol conditions because the results will tell us which SIP process is more important in a polluted and clean environment, and can deepen our understanding of the difference between maritime and continental thunderstorms.

We agree it is still a question that which process of SIP actually plays a role in clouds and whether they all play a role at the same time. In fact, this is one of the purposes of this study. Our results suggest the RS process is the most important one in an environment with high aerosol concentration, and the SD process is more important when the aerosol concentration is low. This conclusion is the same no matter whether all the four SIP processes are considered or individual SIP process is implemented in the model. However, the charging rate is the greatest when all the four SIP processes are turned on.

We acknowledge that the conclusions are obtained only from a case study, and some of the results are different from the other cases in previous studies. To illustrate the similarity and difference between this study and previous ones, we add the following discussions in the revised paper:

[revised manuscript text omitted]

2. Regarding the structure of the paper, in the description of the model in section 2.2.2, the content of this section is almost a duplicate of Appendix B in Yang et al. (2024). Is there a need for duplicate descriptions here?

Reply: Thank you for your comment. According to your suggestion, the equations in Section 2.2.2 have been removed, and the related text has been modified.

3. From the results in Fig. 10, compared to the experiments with N0=400, the experiments with N0=4000 show a stronger charging rate, which also corresponds to a stronger electric field intensity in these two groups of experiments (Fig. 16). However, in Fig. 10, compared with noSIP-4000, the non-inductive charging rate of 4SIP-4000 is stronger, but the polarity and height of the charging rate change very little, which may not cause a significant change in the polarity of the charge structure. On the contrary, the inductive charging rate has undergone significant changes. This may be the reason for the change in the polarity of the charge structure of 4SIP-4000 in Fig. 13.

Reply: Thank you for your comment, and sorry for the confusing statement in the original manuscript. As shown in Fig. 11 in the revised manuscript, the charge structure, especially the reversal temperature, can be significantly altered by increasing CCN concentration. Therefore, the modified charge structure in 4SIP-4000 experiment is not only a result of SIP but also the increased CCN concentration. In fact, the reversal temperature is less affected by SIP because the polarity and height of the charging rate change very little by comparing the noSIP-4000 and 4SIP-4000 experiments. Therefore, it is incorrect to state the SIP processes can change the structure, it is the combined effect of increased CCN concentration and SIP processes that modify the charge structure.

The magnitude of noninductive charging rate is much larger than that of inductive charging, so we believe it is still the noninductive charging dominates the changes in total charge structure. To confirm this, we made a sensitivity test using noninductive charging only (Fig. R4), it is seen from the figure that the modelled charge structure is

similar compared to that in the original paper, which included both charging mechanisms.

[Figure]

Figure R4. Time-height evolution of mean charge carried by (a) graupel/hail (a-d) and (b) ice/snow particles as well as (c) mean total space charge (unit is nC m⁻³) from the simulation 4SIP-4000 with noninductive charging only.

Many previous studies concluded that noninductive charging process is the main charging process of thunderstorms and the inductive charging alone would be insufficient to strongly electrify a storm (Brooks and Saunders, 1994; Jayaratne et al., 1983; Mansell et al., 2005; Saunders and Peck, 1998; Takahashi, 1978). This paper also provides the same conclusion that the noninductive charging rate is much greater than the inductive charging rate (shown in Fig. 11 in the revised manuscript). In addition, the Fig. 15 in Yang et al. (2024) shows the graupel charge density and noninductive charging rate in noSIP experiment and RS experiment with only noninductive charging considered. In Fig. 15 in Yang et al. (2024), the distribution of graupel charge density still produces a significant variation. We agree that the inductive charging rate has undergone significant changes, so the effect of induced charging rates is also of concern. Based on the above discussion, it can be concluded that both noninductive charging process and inductive charging process have impact on charge structure, and the former has a greater impact.

References:

Brooks, I. M. and Saunders, C. P. R.: An experimental investigation of the inductive mechanism of thunderstorm electrification, J. Geophys. Res., 99, 10627, https://doi.org/10.1029/93JD01574, 1994.

Jayaratne, E. R., Saunders, C. P. R., and Hallett, J.: Laboratory studies of the charging of soft-hail during ice crystal interactions, Q.J Royal Met. Soc., 109, 609–630,

https://doi.org/10.1002/qj.49710946111, 1983.

Mansell, E. R., MacGorman, D. R., Ziegler, C. L., and Straka, J. M.: Charge structure and lightning sensitivity in a simulated multicell thunderstorm, J. Geophys. Res., 110, D12101, https://doi.org/10.1029/2004JD005287, 2005.

Saunders, C. P. R. and Peck, S. L.: Laboratory studies of the influence of the rime accretion rate on charge transfer during crystal/graupel collisions, J. Geophys. Res., 103, 13949–13956, https://doi.org/10.1029/97JD02644, 1998.

Takahashi: Riming Electrification as a Charge Generation Mechanism in Thunderstorms, J. Atmos. Sci., 35, 1536–1548, https://doi.org/10.1175/1520-0469(1978)035<1536:REAACG>2.0.CO;2, 1978.

Yang, J., Huang, S., Yang, T., Zhang, Q., Deng, Y., and Liu, Y.: Impact of ice multiplication on the cloud electrification of a cold-season thunderstorm: a numerical case study, Atmos. Chem. Phys., 24, 5989–6010, https://doi.org/10.5194/acp-24-5989-2024, 2024.

4.  The paper mentions that without SIP, the aerosol does not change its charge structure. However, as shown in Fig. 11a, even without the SIP process, the charging rate within the cloud significantly changes with the change in aerosol. The height-time variation diagram may not reflect the actual charge structure, and a cross-section diagram of the charge structure should be provided.

Reply: Thank you for your comment, and we are sorry for the incorrect statement. We agree that the aerosol concentration strongly affects the charge structure, and in fact, it has a stronger impact than the SIP on the reversal temperature and height. This can be readily seen from the time-height diagram in the original manuscript. In the revised paper, we have modified the statement and showed that the charge structure can be significantly affected by increasing aerosol concentration. According to your suggestion, the cross-sections of total charge density from the noSIP experiment and the 4SIP experiment are shown below. The cross-sections show a more complicated charge structure, weak negative charge is found in some areas near the cloud top in the 4SIP-4000 experiment, but in general, it is consistent with the time-height diagram and demonstrates that the increase in aerosol concentration can strongly affect the charge structure.

[Figure]

Figure R5. The cross-section diagrams of the charge structure from (a-b) noSIP experiment and (c-d) 4SIP experiment.

**Reply to the comments of Reviewer 3**

**General comments:**

Huang et al. present a detailed investigation into the role of secondary ice production (SIP) processes and aerosols in cloud electrification within thunderstorms. Utilizing the Weather Research and Forecasting (WRF) model with a spectral bin microphysics (SBM) scheme, this study examines how different SIP processes interact with varying cloud condensation nuclei (CCN) concentrations to influence cloud microphysics and charge separation. This topic is both timely and relevant, exploring key uncertainties in our understanding of cloud microphysics and electrification mechanisms under different aerosol conditions. Nevertheless, the manuscript requires substantial revisions to improve clarity and strengthen the validity of conclusions. The following specific comments and technical corrections should be addressed before the paper is considered for publication:

Reply: We appreciate your insightful comments. The paper has been revised accordingly and has been improved a lot. Please see our responses below.

**Specific comments:**

1. **In the abstract:** (i) Please be more precise when referring to terms like "clean" and "polluted" environments, as well as "relatively warm/cold temperatures," both in the abstract and throughout the manuscript. Explicitly define these terms or specify the aerosol conditions and temperature ranges you are referring to each time to avoid ambiguity. (ii) It would be beneficial to mention some implications of your findings within the abstract. For example, you could explain the importance of correctly capturing the charging reversal at different temperatures due to variations in aerosol concentrations or the inclusion of SIP. This would help readers understand the relevance and impact of your results. (iii) Finally, please define what is meant by the term "normal charge structure" (Line 29).

Reply: Thank you for your comment. (i) The aerosol conditions and temperature ranges have been clearly described to avoid ambiguity. (ii) The charging reversal at different temperatures is important because it affects the polarity of the charge acquired by the ice-phase particles during charge separation, which in turn affects the charge structure and the flash rate in the cloud. According to the analysis in this paper, it is known that the reversal temperature is affected by both aerosol concentration and SIP processes. It implies the importance of aerosol and SIP processes for the charge structure and discharge process in clouds. (iii) The term "normal charge structure" is defined as "with an upper positive charge region and a lower negative region" in the abstract.

2. **Lines 41-52:** Here you mention the aerosol indirect effects in liquid clouds (cloud albedo and lifetime effect). Consider also discussing the indirect aerosol effects in

mixed-phase clouds (MPCs; e.g., glaciation, riming and thermodynamic indirect effect in convective clouds). This is particularly relevant to your study, and these indirect effects could help explain some of the results you present (e.g., Lines 241-247 or 296-298).

Reply: Thank you for your comment. The indirect aerosol effects in mixed-phase clouds is discussed as follows: "*Aerosol also has significant impacts on the microphysics in mixed-phase region. According to previous studies, the increase in aerosol concentration reduces cloud droplet size, thus more small droplets can be lofted to mixed-phase region, which may enhance the freezing of cloud droplets and facilitate cold rain process (Rosenfeld and Woodley, 2000; Rosenfeld et al., 2008; Hoose et al., 2010; Sherwood, 2002; Jiang et al., 2008). The increase in CCN condensation may also intensify the hail growth through riming in convective clouds (Chen e al., 2010), and the enhanced ice growth rate may produce more latent heat, which in turn invigorate the convections (Khain et al., 2005; Wang, 2005).*".


Reply: Thank you for your valuable comment. Despite the deviation between the observed and simulated values, the simulation results well reproduce the macro-morphology, the occurrence location, and the eastward tendency of this squall line. The observed result is the same as Huang et al. (2024). For the southeastern coast of China, both FNL (Guo et al., 2017; Qu et al., 2017) and ERA5 (Gong et al., 2024; Wang et al., 2023) data are widely used in numerical simulations. Based on your suggestion, we compare the simulated radar reflectivity of these two data. Figure R6 shows the simulated radar reflectivity from the 4SIP-4000 experiment using ERA5 reanalysis data. It is noted that simulation results obtained for FNL and ERA5 data are similar. Therefore, FNL data is kept for use in this paper.

[Figure]

Figure R6. (a) The simulated radar reflectivity from 4SIP-4000 experiment with ERA5 reanalysis data. (b) The observed radar reflectivity.

Table R1. The difference in mean rain mixing ratio and number concentration between noSIP experiment and SD experiment (SD-noSIP) at 0°C to -10°C.

| $N_0$ | Mixing ratio | Number concentration |
|---|---|---|
| 400 | -60.57% | -43.57% |
| 1000 | -75.34% | -62.31% |
| 2000 | -99.42% | -75.87% |
| 4000 | -59.88% | -37.99% |

14. **Figure 6:** Confirm whether the mean vertical profiles in this figure correspond to the same period shown in Figure 5. Additionally, have you tried to use logarithmic scale for the horizontal axes showing the number concentration of particles in this figure?

Reply: Thank you for your comment. These two figures in the revised manuscript share the same period. Figure R7 uses the logarithmic scale on the horizontal axes. In our opinion, the linear scale in the original figure better reveals the difference among different experiments. For example, the difference in graupel number concentration among the four experiments is more pronounced using the liner scale axis.

[Figure]

Figure R7. The vertical profiles of mean concentration for (a) cloud drop, (b) rain, (c) graupel/hail and (d) ice/snow from noSIP-400, noSIP-4000, 4SIP-400 and 4SIP-4000 experiment. The horizontal axes are logarithmic scale.

15. **Figure 7:** This figure appears overly complex, potentially obscuring key insights from your sensitivity experiments. Please consider reducing the number of sensitivities discussed in the main paper or discussing any plateau or asymptotic behavior as CCN concentrations increase. A logarithmic vertical axis might also be beneficial in this figure. Is there an issue with the y-axis in Figure 7b?

Reply: Thank you for your comment. According to your suggestion, the figure is revised and the y-axis in (b) is modified. Now only noSIP and 4SIP results are shown in the figure (Fig. R8). In fact, we only discussed noSIP and 4SIP results in the original paper, so the text remains similar. We tried using logarithmic vertical axis but in our opinion the linear scale in the original figure better reveals the difference among different experiments.

[Figure]

Figure R8. The time-domain-averaged mixing ratios (a, c, e, g) and number concentrations (b, d, f, h) of (a, b) cloud drop, (c, d) rain, (e, f) graupel/hail and (g, h) ice/snow from 8 sensitivity experiments.

[Figure]

Figure R9. The same as Fig. R3, but the vertical axis is logarithmic.

16. **Line 285-286:** Have you examined how SIP rates vary with changes in CCN concentrations?

Reply: Thank you for your comment. Yes, the secondary ice production rate of four SIP processes has been discussed in Fig. 9 (Fig. R10 here) in the original manuscript.

[Figure]

Figure R10. Time-height revolution of secondary ice production rate of four mechanisms of SIP. (a-d) rime-splintering process (RS); (e-h) ice-ice collisional breakup (IC); (i-l) freezing drop shattering (SD) and (m-p) sublimational breakup (SK). The first to fourth column indicate $N_0$=400, 1000, 2000, 4000 cm$^{-3}$, respectively.

17. **Line 313 and Figure 9:** Please elaborate on why IC is most significant within the -10°C to -20°C temperature range. The enhanced efficiency of IC within the so-called dendritic growth layer is discussed in von Terzi et al. (2022) and Georgakaki et al. (2024). Did you implement both IC formulations from Table 1 of Phillips et al. (2017) for dendritic and spatial planar ice crystal habits? If so, was the selection of ice habit based on temperature? This should be further clarified in Section 2.2.1, as it might help explain the results discussed here. Also, please consider using a logarithmic scale for the colorbar in this figure.

Reply: Thank you for your comment. As shown in Von Terzi et al. (2022) and Georgakaki et al. (2024), the collision of dendritic ice is the reason for the enhanced IC process between -10°C and -20°C. The greater ice number concentration between -10°C and -20°C favors the ice collision, while the smaller size of ice above -20°C isotherm is unfavorable for the collision between the ice.

Yes, we do implement both IC formulations from Table 1 of Phillips et al. (2017) for dendritic and spatial planar ice crystal habits. Since the fast-SBM does not separate different ice habits, the selection of ice habits is simply based on temperature. Ice is assumed to be dendritic between -12°C and -17°C and is spatial planar ice between -40°C and -17°C as well as between -9°C and -12°C.

Figure R11 is the same as Fig. 9 in the original manuscript, but the colorbars have

logarithmic scales. In our opinion, the main difference in production rates is clearer with the liner colobar axis than logarithmic colobar.

[Figure]

Figure R11. Time-height evolution of secondary ice production rate of four mechanisms of SIP. (a-d) rime-splintering process (RS); (e-h) ice-ice collisional breakup (IC); (i-l) freezing drop shattering (SD) and (m-p) sublimational breakup (SK). The first to fourth columns indicate $N_0$=400, 1000, 2000, 4000 cm$^{-3}$, respectively. The colorbars have logarithmic scales.


Reply: Thank you for your comment. Since the temperature contours in the time-height plot are obtained by taking the mean value, there is inevitably a bias. In order to better show the distribution of production rates, the cross-section plot is shown. As can be clearly seen in Figure R12, the generation rate is restricted to between -3°C and -8°C.

[Figure]

Figure R12. The cross-section of the production rate of the rime splintering process. (a) $N_0$=400 cm$^{-3}$; (b) $N_0$=1000 cm$^{-3}$; (c) $N_0$=2000 cm$^{-3}$; (d) $N_0$=4000 cm$^{-3}$;

19. **Section 3.3:** Is there any way to evaluate the modeling results and conclusions from this section? In your previous work (Yang et al., 2024), you included flashing rates observations. Were such observations available for the squall line discussed in this study?

Reply: Thank you for your comment. We are sorry for not having observed flash rate data using ground-based stations. However, the comparison of observed (from FY-4A satellite) and simulated lightning locations is shown in Figure 5 (Fig. R13 here) in the revised manuscript. The describes is as follows: "*In this paper, the observed lightning data are collected by the Lightning Mapping Imager (LMI), which is mounted on the second-generation Chinese geostationary meteorological satellite FY-4A to continuously detect lightning in China and its neighbouring areas. Additionally, the brightness temperature captured by FY4A satellite is used to delineate the area of the deep convective cloud. Figure 5 shows the location of the lightning events observed by LMI and simulated in 4SIP-4000 experiment. It is evident that the simulated and observed lightning locations are in good agreement with each other and are both in the*

*low brightness temperature region, which characterizes the presence of strong convection.* ”

[Figure]

Figure R13. The simulated and observed lightning locations (sign “+”). The shaded field indicates brightness temperature.

20. **Lines 435-437:** Could you clarify the need to alter the "inverted tripole structure" of the total charge density? What is meant by "normal charge structure," and to which subplot in Figure 15 does this refer?

Reply: Thank you for your valuable comment. The normal charge structure has an upper positive charge region and a lower negative region and is shown in Fig. 16n and d in the original manuscript. Although we do not have observational data on the charge structure, the normal charge structure is more frequently observed in south China than inverted structure as shown by previous studies (Zhang et al., 1997).

Reply: Thank you for your comment. The "concertation" has been changed to "concentration" in the revised manuscript.

---

## Author Response (AR2)

Dear Editor,

We would like to thank the editor and reviewers for your professional comments and suggestions, which are very helpful in improving our manuscript. We have implemented the technical corrections in the revised paper, and the zenodo dataset is now fully accessible. Our point-by-point response to the comments raised by reviewer is shown as follows. The comments are in black, and our responses are given directly afterward in blue.

**Reply to the comments of Reviewer**

I would like to thank the authors for thoroughly addressing the comments raised by all three reviewers. The revised manuscript has been significantly improved. I have only a few minor comments on this version of the manuscript:

• To improve clarity and organization, please consider moving the description of the observational datasets (reflectivity and lightning locations) and their associated uncertainties from the Results Section (3.1 Model validation) to Section 2, which is dedicated to the description of data and methodology.

Reply: We appreciate your comment. The description of the observational datasets is moved to Section 2.3.

• Lines 137-139: The initial sublimation break-up parameterization has two slightly different formulations—one suited to dendritic snow and the other to graupel particles. Am I correct in understanding that you used only the formulation appropriate for ice/snow particles and did not apply the one suited for high-density solid particles (graupel/hail)? Please clarify this here.

Reply: We appreciate your comment. Yes, we only considered the breakup of ice/snow collision and did not apply the one suited for high-density solid particles. This is now clarified in the paper.

• Line 237: Please specify in the revised manuscript the threshold ($10^{-6}$ g/kg) used to define "in-cloud" conditions, as indicated in your response to the reviewer document.

Reply: We appreciate your comment. This is now clarified in the paper.

• In Lines 465-469: Please update the reference from Fig. 18 to Fig. 17. Additionally, it would be helpful to provide a quantitative estimate for the "remarkably enhanced" flash rate with higher aerosol concentrations or SIP inclusion. For instance, you can mention the maximum factor of increase (e.g., up to tenfold) with increasing aerosol

levels. It also appears that the flash rate reaches a "plateau" beyond an aerosol concentration of approximately 1000 cm⁻³, as there seems to be little difference between the 2000 and 4000 cm⁻³ cases.

Reply: We appreciate your comment. The reference is updated from Fig. 18 to Fig. 17. In addition, the following discussion is added: *"Without SIP processes, the flash rate in the case with a CCN concentration of 4000 cm$^{-3}$ is 2-3 orders of magnitude greater than that with a CCN concentration of 400 cm-3. The flash rate reaches a "plateau" beyond a CCN concentration of approximately 1000 cm$^{-3}$, as there is little difference between the noSIP-2000 and noSIP-4000 cases. The addition of SIP processes can also slightly enhance the flash rate. For the cases with a CCN concentration of 400 cm$^{-3}$ and 4000 cm$^{-3}$, the average flash rates between 16:00 May 29th and 04:00 May 30th increase by 66.8% and 44.3% due to the SIP processes, respectively. Therefore, the flash rate from the 4SIP-4000 experiment is the largest."*